# High quality genomes produced from single MinION flow cells clarify polyploid and demographic histories of critically endangered *Fraxinus* (ash) species

Steven J. Fleck [1✉], Crystal Tomlin[1], Flavio Augusto da Silva Coelho [1], Michaela Richter [1], Erik S. Danielson[2], Nathan Backenstose [1], Trevor Krabbenhoft [1], Charlotte Lindqvist [1] & Victor A. Albert [1✉]

With populations of threatened and endangered species declining worldwide, efforts are being made to generate high quality genomic records of these species before they are lost forever. Here, we demonstrate that data from single Oxford Nanopore Technologies (ONT) MinION flow cells can, even in the absence of highly accurate short DNA-read polishing, produce high quality de novo plant genome assemblies adequate for downstream analyses, such as synteny and ploidy evaluations, paleodemographic analyses, and phylogenomics. This study focuses on three North American ash tree species in the genus *Fraxinus* (Oleaceae) that were recently added to the International Union for Conservation of Nature (IUCN) Red List as critically endangered. Our results support a hexaploidy event at the base of the Oleaceae as well as a subsequent whole genome duplication shared by *Syringa*, *Osmanthus*, *Olea, and Fraxinus*. Finally, we demonstrate the use of ONT long-read sequencing data to reveal patterns in demographic history.

[1] Department of Biological Sciences, University at Buffalo, Buffalo, NY 14260, USA. [2] Western New York Land Conservancy, East Aurora, NY 14052, USA.
✉email: sjfleck@buffalo.edu; vaalbert@buffalo.edu

With populations of threatened and endangered organisms continually declining, it is vital that high quality genomic records of these species are preserved before they are lost forever. In 2017, five North American ash trees (*Fraxinus* L.; Oleaceae) were added to the International Union for Conservation of Nature (IUCN) Red List as critically endangered: white ash (*Fraxinus americana* L.)[1], black ash (*F. nigra* Marsh.)[2], green ash (*F. pennsylvanica* Marsh.)[3], pumpkin ash (*F. profunda* Bush)[4], and blue ash (*F. quadrangulata* Michx.)[5]. These North American ash trees have become critically endangered due to the introduction of the emerald ash borer (EAB, *Agrilus planipennis* Fairmaire), a buprestid beetle[6]. The EAB was first identified in southeast Michigan in 2002 and has since spread to most of the United States, resulting in the death of millions of ash trees as well as massive economic and ecological impacts[6]. The larval stage of the EAB feeds on the phloem of ash trees, girdling them as adults, or even long before the tree reaches maturity (as small as 2.5 cm diameter at breast height), which kills the tree within 2–4 years of infestation[6]. North American ash species have shown some low level of resistance[7], but *F. americana*, *F. nigra*, and *F. pennsylvanica* are highly vulnerable to the EAB, with some studies showing virtually 100% mortality and newly germinated ash seedlings scarce or completely absent[8].

Recent progress has been made on the evolution of EAB resistance in *Fraxinus*. Kelly et al.[7] identified fifty-three candidate *Fraxinus* genes for EAB resistance and demonstrated that forty-eight likely arose through convergent evolution in three separate lineages (taxonomic sections) within the genus: *F. mandshurica* Rupr., a member of the *Fraxinus* section; *F. platypoda* Oliv., sister to the *Melioides* section; *F. baroniana* Diels, *F. floribunda* Wall., and *Fraxinus sp.* D2006-0159, clustered within section *Ornus*. All five EAB-resistant species are native to Asia and within the native range of the EAB, except for the present-day ranges of *F. baroniana* and *F. floribunda*. Loss-of-function mutations for candidate genes were more common in ash trees that were susceptible to the EAB than species that were resistant[7]. Out of the fifty-three candidate genes, twenty-four have putative roles relating to defense response against insect herbivory, including phytohormone biosynthesis and signaling, and others have potential roles related to defense response.

Recent progress has also been made in comparative genomics of the Oleaceae. High quality, chromosome-level genome assemblies have been published for Arabian jasmine (*Jasminum sambac* (L.) Aiton)[9], sweet osmanthus (*Osmanthus fragrans* Lour. var. 'Rixianggui')[10] and the cultivar *O. fragrans* 'Liuyejingui'[11]), weeping forsythia (*Forsythia suspensa* (Thunb.) Vahl)[12,13], common olive (*Olea europaea* L.)[14], and early lilac (*Syringa oblata* Lindl.)[15,16]. For the genus *Fraxinus*, Kelly et al.[7] published twenty-eight short-read de novo assemblies, representing twenty-two species-level taxa, including three *F. pennsylvanica* assemblies, one EAB-susceptible (pe-48), one partially EAB-resistant (pe-00248), and one originally misidentified as *F. caroliniana*. Additionally, three *F. angustifolia* Vahl subspecies and two unidentified *Fraxinus* species were sequenced[7]. Huff et al.[17] generated new 800 basepair (bp) insert size library data for eight of these individuals, as well as for an additional species, *F. apertisquamifera* H.Hara, and created new short-read de novo assemblies. They also generated a chromosome-level assembly for a rare, putatively EAB-resistant *F. pennsylvanica* individual. These twenty-six short-read assemblies, along with the previously published *F. excelsior* L[18]., were scaffolded using the *F. pennsylvanica* reference assembly as a guide[19]. The resulting pseudo-chromosome-level assemblies were able to place about 45%–87% of the total lengths of the short-read input assemblies into pseudomolecules.

Polyploidy events have been considered important for the evolutionary radiation of the Oleaceae. Within the genome of *Jasminum sambac*, roughly 30% of genes in rapidly expanding gene families were associated with a whole genome hexaploidy event (sometimes called a triplication or three-subgenome structured event)[9]. These adaptive hypotheses involve duplicates of numerous stress-related and secondary metabolism associated genes, with the evolution of floral fragrance thought to be influenced by polyploidization and tandem duplication. It is possible for polyploidy events to increase diversification rates and species richness[20], and the proposed hexaploidy event at the base of the Oleaceae may have contributed to the diversification and adaptive potential of the Oleaceae since its divergence from other Lamiales[9]. Likewise, a more recent whole genome duplication (WGD) event shared by all members of the Oleeae tribe may have been important for the evolution of these species. In *Osmanthus fragrans* 'Rixianggui', WGD and tandem gene duplication were hypothesized to have a role in the evolution of the monoterpene synthesis pathway that generates the strong, sweet aroma of its flowers[10]. Additionally, the functional divergence of oil biosynthesis pathway genes following duplication may have been important in the evolution of *Olea europaea* var. *sylvestris*, an important oil crop[21].

Here, we generate highly accurate and coding sequence-complete haploid genome assemblies for the critically endangered species white ash (*Fraxinus americana*), black ash (*F. nigra*), and green ash (*F. pennsylvanica*) as part of the Org.ONE project. Org.ONE is a pilot-stage project initiated by Oxford Nanopore Technologies (ONT) that aims to support "equitable, faster, and more localized" ultra-long read DNA sequencing and subsequent de novo genome assemblies of critically endangered species (https://org.one/oo). Org.ONE's goal is to support local researchers to collect and sequence critically endangered species close to the sample's origin instead of shipping samples to centralized locations for sequencing and assembly. All sequencing data for critically endangered species is uploaded to the EBI public database (https://www.ebi.ac.uk/ena/browser/home) for anyone to use to help guide conservation efforts. This data can then be used to generate reference assemblies that may provide insights into the genome architecture of a species, the genetic diversity within and between populations, demographic history and effective population sizes, hybridization and introgression between species, and identify deleterious mutations as well as local adaptation on the genomic scale[22]. For example, identifying locally adapted variants can be useful for introducing genetic diversity to populations with higher levels of inbreeding. While chromosome-level assemblies are ideal for reference genomes, in the case of the Tasmanian devil (*Sarcophilus harrisii*), a much less contiguous reference genome (35,974 scaffolds, N50 1.85 Mbp) has had major impacts in conservation management of that species[23]. We therefore consider our own cost- and time-effective work here to provide a justifiably valuable conservation resource within the plant genomics arena.

The three ash species were sequenced using a single ONT MinION flow cell each. Advancements in homology-based gene prediction allowed us to produce high quality genome annotations without needing to produce RNA-seq reads for our samples. While our genome assemblies are not chromosome-level, they are improvements over published short-read assemblies in regard to their sequence completeness and contiguity, haploid assembly size, and the capacity to scaffold the primary assemblies into pseudo-chromosomes using a reference. The resulting reads and genome assemblies were of sufficient accuracy to evaluate polyploidy level and paleodemographic history using long-read instead of the deep short-read sequence data typically employed. While this study focuses on relatively small genome assemblies (1 C < 1000 megabases), the methods we describe can be applied to larger genomes with further

**Table 1 Assembly and annotation statistics for *Fraxinus* species after running Purge Haplotigs.**

| Assembly | *F. americana* | *F. nigra* | *F. pennsylvanica* | *F. pennsylvanica* v1.4 |
|---|---|---|---|---|
| # contigs | 4364 | 2544 | 6764 | 110 |
| Largest contig | 3,688,385 | 6,590,250 | 6,486,905 | 56,547,140 |
| Est. Total length | 875 Mbp | 829 Mbp | 869 Mbp | 869 Mbp |
| Total length | 851,858,478 | 776,258,169 | 841,639,580 | 756,791,283 |
| GC (%) | 35.26 | 34.76 | 35.2 | 34.40% |
| N50 | 507,263 | 1,081,099 | 244,798 | 33,221,578 |
| L50 | 450 | 206 | 928 | 10 |
| # N's per 100 kbp | 0.80 | 0.48 | 0.62 | 12,120.86 |
| Complete BUSCOs | 1561 (96.7%) | 1561 (96.7%) | 1553 (96.2%) | 1576 (97.6%) |
| Complete single-copy BUSCOs | 1281 (79.4%) | 1302 (80.7%) | 1271 (78.7%) | 1308 (81.0%) |
| Complete duplicated BUSCOs | 280 (17.3%) | 259 (16.0%) | 282 (17.5%) | 268 (16.6%) |
| Fragmented BUSCOs | 37 (2.3%) | 25 (1.5%) | 37 (2.3%) | 26 (1.6%) |
| Missing BUSCOs | 16 (1.0%) | 28 (1.8%) | 24 (1.5%) | 12 (0.8%) |
| Total BUSCOs searched | 1614 | 1614 | 1614 | 1614 |
| **Annotation** | ***F. americana*** | ***F. nigra*** | ***F. pennsylvanica*** | ***F. pennsylvanica* v1.4** |
| gene model/mRNA count | 41,093/46,464 | 37,496/42,777 | 40,446/45,696 | 35,470/35,470 |
| Complete BUSCOs | 96.9% (1564) | 97.3% (1571) | 97.6% (1575) | 82.5% (1332) |
| Complete single-copy BUSCOs | 78.7% (1271) | 80.3% (1296) | 79.4% (1281) | 70.8% (1142) |
| Complete duplicated BUSCOs | 18.2% (293) | 17.0% (275) | 18.2% (294) | 11.8% (190) |
| Fragmented BUSCOs | 0.9% (15) | 0.9% (14) | 1.0% (16) | 2.4% (38) |
| Missing BUSCOs | 2.2% (35) | 1.8% (29) | 1.4% (23) | 15.1% (244) |
| Total BUSCOs searched | 1614 | 1614 | 1614 | 1614 |

Species names are italicized.
*F. pennsylvanica* v1.4 is the reference assembly from Huff et al.[17]

improvements in sequencing outputs from MinION (or higher output PromethION) ONT flow cells.

## Results and discussion

**Genome assembly and annotation.** Totals of 6,065,809, 3,788,248 and 2,581,752 reads with N50s of 10,916, 11,213, and 34,165 were generated for wild-collected, Chautauqua County, New York *Fraxinus americana*, *F. nigra*, and *F. pennsylvanica* individuals, respectively (Table S1; Fig. S1). Mean 1C value estimates for *F. americana*, *F. nigra*, and *F. pennsylvanica* were 0.895 pg (875 Mbp), 0.847 pg (829 Mbp), and 0.889 pg (869 Mbp), as calculated using data from Whittemore et al.[24] Comparing estimated genome sizes with total bases generated, *F. americana*, *F. nigra*, and *F. pennsylvanica* had 25.0x, 22.4x, and 23.0x coverage, respectively.

The initial de novo genome assemblies for *Fraxinus americana*, *F. nigra*, and *F. pennsylvanica* had high levels of completeness based on Benchmarking Universal Single-Copy Orthologs (BUSCOs)[25], i.e., 97.0%, 96.7%, and 97.0% complete BUSCOs from the embryophyta_odb10 database, respectively (Table S2). These completeness levels were very similar to the unannotated *F. pennsylvanica* reference assembly (v1.4) from Huff et al.[17] with 97.6% complete embryophyta_odb10 BUSCOs (Table S2). Repetitive elements made up nearly 60% of all three *Fraxinus* genome assemblies (Table S3), while they made up about only about 45% of the *F. pennsylvanica* reference assembly, which was based on Illumina short reads. Since short-read assemblies have the propensity to collapse duplicated sequences, this difference was likely due to better read-through of repetitive DNA on long ONT single molecules[26]. The largest group of retroelements discovered were Ty1/Copia and Gypsy/DIRS LTR elements, each comprising about 15% of all repetitive elements characterized.

Annotations were generated using GeMoMa[27,28], a homology-based gene prediction software, using annotations from the *Fraxinus pennsylvanica* reference assembly[17], *Fraxinus excelsior*[18], *Olea europaea*[14], *Jasminum sambac*[9], and *Arabidopsis thaliana*[29]. Our resulting *F. americana*, *F. nigra*, and

*F. pennsylvanica* annotations contained 49,500, 38,374, and 50,130 gene models, encompassing 97.4%, 97.4%, and 98.1% complete BUSCOs, respectively (Table 1), each about 15% more than the chromosome-level *F. pennsylvanica* assembly[17] at 82.5% complete BUSCOs.

Our initial genome assemblies for *Fraxinus americana* and *F. pennsylvanica* contained many uncollapsed haplotypic regions (Fig. S2a, c, e), suggesting that our samples were highly heterozygous[30]. Using HapPy[30], *F. americana* and *F. pennsylvanica* were shown to have a haploid peak around twenty, with a diploid peak around ten, representing uncollapsed diploid sequences. *F. nigra* only had a single coverage peak around ten, likely resulting from low-quality quality scores for that sample's ONT reads that were filtered from the analysis. HapPy also measured that the *F. americana*, *F. nigra*, and *F. pennsylvanica* assemblies were 75.8%, 99.6%, and 66.0% haploid, respectively (Table S4). This was corroborated by greater-than-expected gene model prediction numbers and duplicated BUSCO counts compared with the *F. pennsylvanica* reference (35,470 gene models) in these initial assemblies, as well as larger than expected haploid assembly sizes, particularly for *F. americana* and *F. pennsylvanica* (Table S2). Lastly, self-vs-self frequency plots of synonymous substitution rates (Ks) between ostensibly syntenic duplicate gene pairs indicated uncollapsed haplotigs within the assemblies as peaks with low Ks that did not correspond to WGD events (Fig. S3d, e, f).

We therefore generated largely haploid assemblies by post-processing using Purge Haplotigs[31], resulting in similarly high levels of complete BUSCOs of 96.7%, 96.7%, and 96.2% in *Fraxinus americana*, *F. nigra*, and *F. pennsylvanica*, respectively (Table 1). HapPy detected only haploid peaks for all three purged assemblies (Fig. S2b, d, f), and self-vs-self Ks frequency plots between syntenic coding sequence (CDS) pairs indicated that diploid sequences had largely been removed from these assemblies (Fig. S4d, e, f). Still, our three *Fraxinus* assemblies, as well as the Huff et al.[17] *F. pennsylvanica* assembly, had high percentages of complete duplicated BUSCOs: 17.3%, 16.0%, 17.5%, and 16.6% for *F. americana*, *F. nigra*, *F. pennsylvanica*, and the *F. pennsylvanica*

reference assembly, respectively. These high levels of duplicated BUSCOs may be a result of the relatively recent WGD in the Oleeae. This pattern is also shared with close relatives, such as *Syringa oblata*[16], *Osmanthus fragrans*[10], and *Olea europaea*[14], which have 10.0%, 20.0%, and 13.3% duplicated BUSCOs, respectively. In the case of *O. fragrans* and *O. europaea*, these assemblies show unpurged heterozygosity in their self-vs-self SynMap Ks plots (Fig. S5). Compared with the *F. pennsylvanica* reference assembly and the reference-guided *F. americana* and *F. nigra* assemblies produced by Huff et al.[17], our assemblies contain slightly more heterozygosity in the form of low frequency $\log_{10}$ Ks values below -1.0 between syntenic paralogous gene pairs. The Huff et al.[17] assemblies are virtually devoid of $\log_{10}$ Ks values below $-1.0$ (Fig. S4g, h, i).

Repeat annotation was executed again on the haplotig-purged assemblies. In all cases, our assemblies identified more Copia and Gypsy LTR elements, DNA transposons, and total interspersed repeats compared to the *F. pennsylvanica* reference assembly (Table S5). The assemblies from this study were also closer to genome sizes estimated by flow cytometry[24], while the reference-guided assemblies were over 100 Mbp smaller than estimated genome sizes. Again, these disparities may be partly due to better repeat detection in long-read genome assemblies, which likely contain fewer collapses of identical or near-identical sequences[26].

In addition to the de novo genome assemblies, we also scaffolded each into pseudo-chromosome-level assemblies using the *F. pennsylvanica* reference assembly as a guide, as was similarly done in Huff et al.[17] for their short-read primary assemblies. We then compared our chromosome-level assemblies generated from ONT reads against the chromosome-level assemblies generated from Illumina short reads from Huff et al.[17]. For this comparison, these de novo genome assemblies were scaffolded using RagTag's scaffolding tool[19]. Since the reference came from a different species or population from the assemblies we were scaffolding, options to do misassembly correction and gap filling were not carried out. It should be noted that *F. nigra* is within section *Fraxinus*, relatively distantly related to *F. americana* and *F. pennsylvanica* within the section *Melioides*. However, the estimated timing of the divergence of the *Fraxinus* and *Melioides* sections is not consistent within the literature. For example, Hinsinger et al.[32] places this event at 44.2 mya, while older events, such as the divergence between *Fraxinus* and *Olea* (21.7 mya;[21] 39.43 mya[33]) and the divergence of *Jasminum* from the rest of the Oleaceae (31.1 mya[34]) have been estimated to have younger ages. Regardless of the age of the divergence between *F. nigra* and the *F. pennsylvanica* reference genome we used, using *F. pennsylvanica* to scaffold a distantly related member of the genus does increase the risk of missing chromosomal rearrangements in the assembly that is reference-scaffolded.

When comparing the ability to scaffold long-read assemblies and short-read assemblies into chromosome-level pseudomolecules, our long-reads did a superior job. Huff et al.[17] was able to scaffold 78.25% and 66.44% of the short-read *F. americana* and *F. nigra* assemblies, respectively, using their *F. pennsylvanica* reference assembly. Using the long-read primary assemblies generated in this study, RagTag was able to scaffold 95.40%, 97.29%, and 95.74% of our *Fraxinus americana*, *F. nigra*, and *F. pennsylvanica* assemblies, respectively (Table S6). Additionally, the scaffolded short-read *F. americana* and *F. nigra* assemblies had 95.04% and 95.91% complete BUSCOs, respectively, and 10.84% and 10.59% duplicated BUSCOs, respectively[17], whereas our scaffolded long-read *F. americana*, *F. nigra*, and *F. pennsylvanica* assemblies found 97.3%, 96.9%, and 96.8% complete BUSCOs, respectively, and 17.5%, 16.2%, and 17.7% duplicated BUSCOs, respectively (Table S7).

**Ploidy analysis.** All three *Fraxinus* species sequenced for this study showed evidence of two polyploidy events following the *gamma* hexaploidy event at the base of all core eudicots (Fig. S4d, e, f). When post-speciation and post-polyploidy gene retention was compared between syntenic blocks of *Fraxinus* and *Vitis vinifera*, which lacks any additional polyploidy events beyond the *gamma* hexaploidy event[35], each *Fraxinus* species showed a six-to-one ratio against *V. vinifera* (Fig. 1b, c, d). On close inspection of the gene retention patterns between *Vitis* and the three *Fraxinus* spp., similar patterns exist in three groups of two, strongly suggesting an additional hexaploidy event in the ancestry of the genus, followed by a single WGD event. Additionally, two pairs of syntenic blocks showed a high extent of post-polyploidy duplicate gene retention, while the final syntenic block pair experienced relatively high duplicate gene loss. While this study only reports the patterns and number of retained genes within these subgenomes, such results might suggest an allopolyploidy event, wherein one subgenome came from one ancestral species, while two subgenomes derived from another, and as such reflecting subgenome dominance patterns as similarly described for other species groups[36,37]. Previously, these (as presently known) Oleaceae-specific polyploidy events were proposed to be two or more WGDs[10,17,21,38,39]. To the best of our knowledge, Xu et al.[9] were the first to propose a secondary hexaploidy event at the base of Oleaceae, about 66 million years ago (Mya). Wang et al.[16] independently corroborated this hexaploidy followed by a WGD for *Syringa oblata*. All currently published members of the tribe Oleeae (i.e., *Osmanthus fragrans*[11], *Olea europaea*[14], and *S. oblata*[16]) showed the same ploidy level as our *Fraxinus* species (Fig. S6).

Wang et al.[16] also identified two subgenomes in *S. oblata*, the result of the most recent WGD event. The duplication that created these subgenomes was estimated to have occurred 28.71 Mya, very close to previous estimates[14,17,18,21] and to the divergence of the *Syringa* genus from *Olea* and *Osmanthus*. The subgenomes had unequal chromosome numbers, with ten chromosomes in subgenome-A and thirteen in subgenome-B. All current Oleeae assemblies show strong macrosynteny with each subgenome of *S. oblata*[16], suggesting shared basic subgenome numbers of ten to thirteen chromosomes (Fig. S6). All five Oleaceae tribes recognized by Wallander and Albert[40] have different basic chromosome numbers: $x = 23$ in Oleeae, $x = 11-13$ in Jasmineae, $x = 14$ in Forsythieae, $x = 13$ in Fontanesieae, and $x = 11$ in Myxopyreae[41,42]. The most thorough phylogenetic analysis to date suggests that Jasmineae and Forsythieae are most closely related to the Oleeae[33]. It seems likely that the ancestor of the Oleeae, Jasmineae, and Forsythieae had a basic chromosome number of thirteen or fourteen, with the ten chromosomes of subgenome-A being derived from a thirteen or fourteen-chromosome ancestor, perhaps via chromosome arm fusion.

*Jasminum sambac*, an early-branching member of the Oleaceae, showed a three-to-one syntenic block ratio with *V. vinifera* (Fig. S7a, d), suggesting that it only contains the hexaploidy event also seen in *Fraxinus*, corroborating the findings of Xu et al.[9] The fractionation bias plot for *Jasminum* against *Vitis* (Fig. S7b, c) agrees with what was observed with *Fraxinus* against *Vitis*: two subgenomes with similar patterns of high gene retention and one subgenome with low gene retention. *Forsythia suspensa*[12,13] showed a one-to-one syntenic block ratio with *Jasminum* (Fig. S8), suggesting that it also contains only the Oleaceae hexaploidy event since its divergence from *V. vinifera*. The exact details of this hexaploidy lacks syntenic evidence outside of the Oleaceae, however, where it remains known only from that family. Xu et al.[9] placed the event at the base of the Oleaceae, but it is unclear if this event occurred before it diverged from its sister family, the

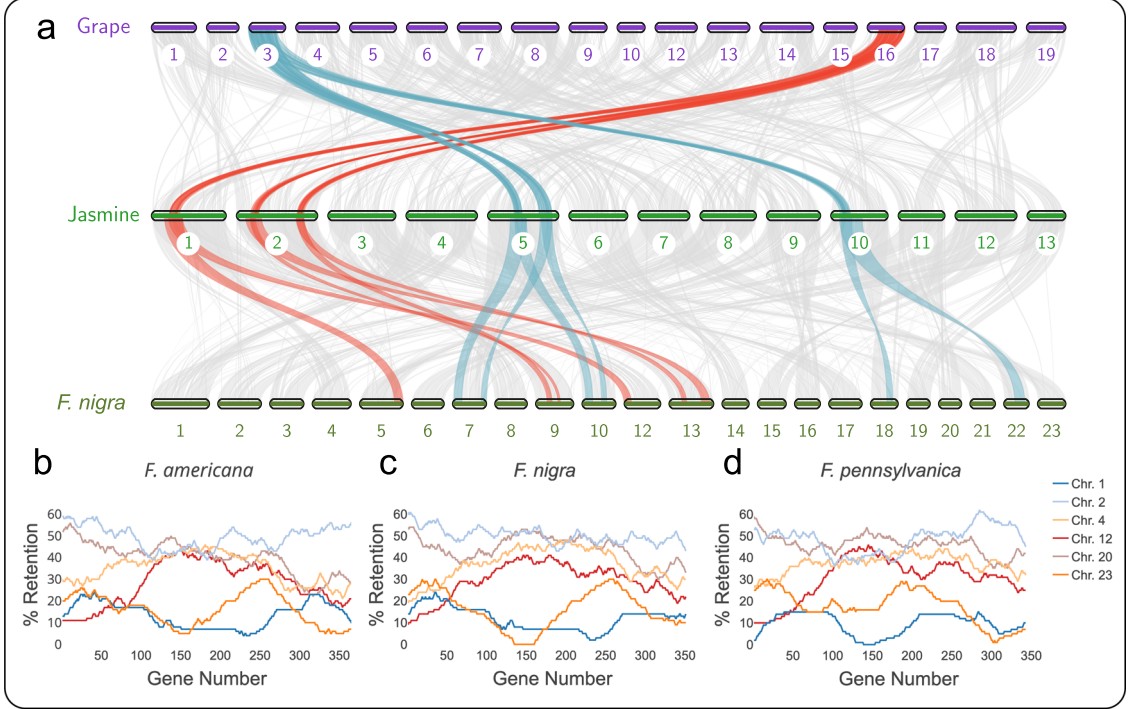

**Fig. 1 Polyploidy events in jasmine and ash since their divergence from grapevine. a** Macrosynteny on a chromosome-scale between grapevine (*Vitis vinifera*) and jasmine (*Jasminum sambac*) and black ash (*Fraxinus nigra*). Numbers below chromosomes indicate chromosome number. Syntenic blocks between *Vitis* chromosome 3, *Jasminum*, and *Fraxinus* are connected by light blue lines. Syntenic blocks between *Vitis* chromosome 16, *Jasminum*, and *Fraxinus* are connected by red lines. Fractionation bias between syntenic blocks of *Vitis* chromosome 19 and *F. americana* (**b**), *F. americana* (**c**), and *F. pennsylvanica* (**d**) reveals 3 pairs of subgenomes (for six subgenomes total), one pair of which retaining far fewer syntenic genes since the ancient hexaploidy at the base of Oleaceae. The Y-axis represents the percent of gene retention of each *Fraxinus* syntenic block compared with windows of genes on *Vitis* chromosome 19. The X-axis represents 100-gene windows along *Vitis* chromosome 19. Line color corresponds with *Fraxinus* chromosome numbers in the key to the right of (**d**).

Carlemanniaceae, as Zhang et al.[38] suggested. Zhang et al.[38] detected an increase in duplicated ortholog groups in a clade containing *Silvianthus bracteatus* (Carlemanniaceae) and eight members of the Oleaceae, but without a published genome assembly to run syntenic analysis with, we will remain cautious in accepting evidence for a shared polyploidy event determined by synonymous substitution (Ks) peaks alone[43,44].

A syntenic dot plot was generated between each long-read *Fraxinus* assembly and *Jasminum sambac*. There were two large syntenic blocks of genes in *Fraxinus* for every one block of genes in *Jasminum* (Fig. S9a), suggesting that a WGD occurred in the ancestor of *Fraxinus* since its divergence from *Jasminum*. When fractionation bias was calculated[45,46] between each long-read *Fraxinus* assembly and *J. sambac*, two *Fraxinus* syntenic blocks displayed similar patterns of gene retention against one chromosome of *J. sambac* (Fig. S9b, c, d). Unlike the sub-genomes that resulted from the hexaploidy event (Fig. S7b, c), these sub-genomes did not show a clear gene retention bias (Fig. S9b, c, d) and may have been the result of an autopolyploid WGD event. Again, this pattern was the same between *Jasminum* and published genomes of other members of Oleeae. Taken together, these results support a hexaploidy that includes at least the Oleeae, Jasmineae, and Forsythieae tribes, followed by a WGD in the ancestor of the Oleeae. Basic chromosome numbers for the Fontanesieae and Myxopyreae suggest the same ploidy level as in Jasmineae and Forsythieae, but syntenic data is lacking for them. Echoing the Ks evidence noted above, it is unclear if chromosome counts in the Carlemanniaceae ($x = 15$ in *Carlemannia* and $x = 19$ in *Silvianthus*[47]) reflect a shared hexaploidy event with Oleaceae, as neither of these counts share the basic chromosome count with the Oleeae stem lineage, which appears to be 11 or 13[41,47].

**Population size history.** Paleodemography for each *Fraxinus* species was determined using an R port for the Pairwise Sequentially Markovian Coalescent (PSMC) model[48]. Mutation rates for the *Fraxinus* genus have not been well established. Previously, Bai et al.[49] used a mutation rate of $2.5 \times 10^{-9}$ substitutions per site per year for both *Fraxinus excelsior* and *Populus trichocarpa*[49]. Additionally, Sollars et al.[18] used a mutation rate of $7.5 \times 10^{-9}$ substitutions per site per year for *F. excelsior*, which was based on a study of *Arabidopsis thaliana*. Looking beyond publications that include *Fraxinus*, Xie et al.[50] measured that peach (*Prunis persica*) had a mutation rate of $7.77 \times 10^{-9}$ substitutions per site per generation; this species has a much lower per-year mutation rate than *A. thaliana* because of its longer generation time, which would also be true for *Fraxinus* spp.. For this study, we therefore used the mutation rate for peach for consistency with a long-lived life cycle.

As noted previously, divergence times within the Oleaceae family and *Fraxinus* genus have not been well established. The coalescence between *F. nigra* within the section *Fraxinus* and *F. americana* and *F. pennsylvanica* within the section *Melioides* is likely too old to be detected with PSMC. Even still, the demographic curves of the three *Fraxinus* spp. in this study had very similar shapes (Fig. S10). While it is unlikely that these curves should coalesce, the demographic curve for *F. nigra* may be shifted toward smaller effective population sizes and toward present day, with the curve shapes potentially reflecting responses

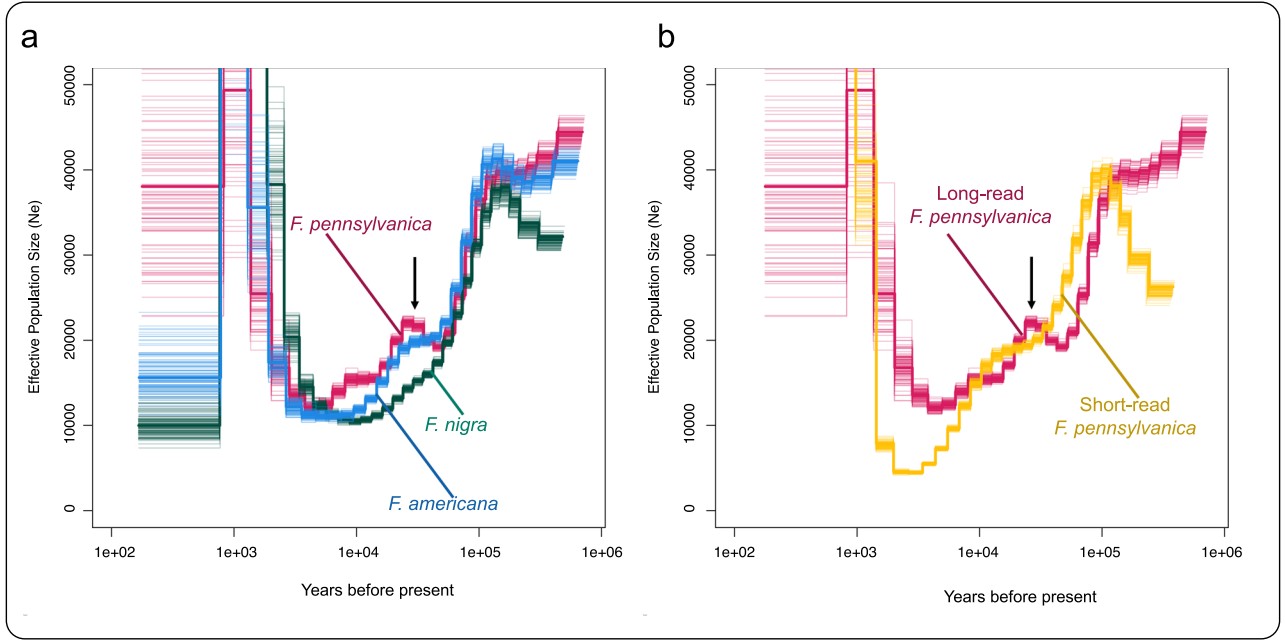

**Fig. 2 Pairwise Sequentially Markovian Coalescent with R (PSMCR) paleodemographic analyses for our long read *Fraxinus* species and the short read *F. pennsylvanica* reference assembly. a** PSMCR for our three long-read *Fraxinus* assemblies. **b** PSMCR for our long-read *F. pennsylvanica* assembly and the short read *F. pennsylvanica* assembly. Demographic curve colors represent the following genome assemblies: blue: *F. americana*; green: *F. nigra*; pink: *F. pennsylvanica*; gold: short-read *F. pennsylvanica*. PSMCR plots generated by mapping *F. americana, F. nigra,* and *F. pennsylvanica* ONT long-reads against their own ONT long-read assemblies. All curves assume a generation time of 15 years. Mutation rates for *F. americana* and *F. nigra* curves have been adjusted in order to correct for differences in segregating sites and genome size. Mutation rates used were $6.76 \times 10^{-9}$, $4.60 \times 10^{-9}$, $7.77 \times 10^{-9}$, and $2.51 \times 10^{-9}$ nucleotide generation$^{-1}$ for *F. americana, F. nigra,* the long-read *F. pennsylvanica,* and the short-read *F. pennsylvanica* respectively. The bcftools multiallelic and rare-variant calling model was used and maxt was set to 10 for PSMCR. The thin black arrow points to a pronounced Ne hump in the long-read *F. pennsylvanica*'s demographic curve.

to a shared geologic history. Curve shifts via heterozygosity differences have been convincingly argued for based on simulated selfing data for different lychee (*Litchi chinensis* Sonn.; Sapindaceae) populations[51]. Here, this same effect would suggest that our *F. nigra* individual may stem from a more inbred population compared with *F. americana* and *F. pennsylvanica*. This observation agrees with the observation that *F. nigra*'s sequenced reads were much more homozygous than the other two species. By artificially lowering the mutation rate for the purposes of comparative analysis, the demographic curve for *F. nigra* overlapped more closely with *F. americana* and *F. pennsylvanica* (Fig. 2a), suggesting that they may have experienced similar patterns of effective population size over the last million years.

We also generated a PSMC curve for the *F. pennsylvanica* reference assembly using its own Illumina short-read data. While it was unlikely we should see a coalescence between any two species, the curves for *F. pennsylvanica* from this study and *F. pennsylvanica* from Huff et al.[17] should coalesce into the same lineage. Similar to our *F. nigra* individual, the population of the *F. pennsylvanica* reference individual appeared to be shifted toward smaller effective population sizes and more recent times (Figs. S10, S11a), suggesting that it may also stem from a more inbred population compared with our own *F. pennsylvanica* accession. Alternatively, there may be an artifact of PSMCs generated from long-read data that spuriously shifts them toward larger effective population sizes and older coalescence times. When the mutation rate was artificially lowered for the short-read *F. pennsylvanica* curve using the same method used on the long-read curves, it did not overlap with our *F. pennsylvanica* like the long-read curves did (Fig. S11b). The mutation rate required additional lowering to overlap with our long-read *F. pennsylvanica* curve.

The PSMC curve for *F. pennsylvanica* from Huff et al. also lacked an upward hump in effective population size ($N_e$) that our *F. pennsylvanica* showed (Fig. 2b). As hypothesized for PSMC analyses in other species groups, this rise in $N_e$ may indicate a change in population heterozygosity through an admixture event[52,53]. Indeed, Huff et al.[17] described their reference assembly as coming from an admixed population, rather than a pure northern or pure southern population. As such, our *F. pennsylvanica* individual may have come from a population with even more admixture between the two populations than the Huff et al.[17] reference accession, which may stem from a population with less admixture than previously thought. Alternatively, *F. pennsylvanica* and *F. americana* have been known to successfully hybridize[54,55] and our individual may come from a population with a history of introgression between the two species. For dating the potential admixture event, it is important to note that an accurate mutation rate and generation time is not known for the *Fraxinus* spp. in this study. While these two values do not affect the shape of the plot, dating of demographic events will not be precise[56,57]. Also possible is that PSMCs generated using ONT long reads are less trustworthy indications of demographic size history, perhaps due to read accuracy differences, or that to the contrary, they actually reveal more runs of heterozygosity useful for demography estimation.

When PSMCs were generated using Illumina short reads from the *F. pennsylvanica* reference mapped onto our *F. pennsylvanica* assembly instead, the $N_e$ hump was no longer visible (Fig. S12a). Likewise, when we aligned our ONT long reads from our *F. pennsylvanica* with the Huff et al.[17] reference assembly, we found that the Ne hump was absent (Fig. S12b). This suggests that our two individuals potentially derive from populations with disparities in their population structure. However, it is also clear

that use of homologous reference assemblies for PSMC analyses is an important factor, even within the same species, especially when assemblies are of different sizes and potentially capture different repetitive element profiles (e.g., Patil & Vijay[58]).

**Phylogenetic orthology inference**. OrthoFinder was run on our assemblies and a selection of proteomes from other high quality Oleaceae, other Lamiales, other asterid, and rosid genome assemblies. A total of 721,170 genes (94.7% of total) were clustered into 37,087 orthogroups, 9801 of which were species specific. An additional 7630 orthogroups contained genes from all 21 species and 200 were single copy orthogroups.

The OrthoFinder species tree grouped the *Fraxinus* assemblies from Huff et al.[17] consistently with the assemblies generated in this study (Fig. S13a). This tree differed from previous phylogenies[33,40,41] by not placing *Jasminum sambac* (Jasmineae tribe) sister to *Osmanthus*, *Olea*, *Syringa*, and *Fraxinus* (Oleeae tribe). Instead, *Forsythia suspensa* (Forsythieae tribe) was sister to the Oleeae, and *Jasminum* was sister to the rest of the Oleaceae. A recent study suggested that phylogenetic discordance in the placement of the five Oleaceae tribes is more likely due to hybridization and ancient widespread introgression, rather than incomplete lineage sorting[33]. Furthermore, the Oleeae may have originated from a hybridization event between the Forsythieae and either the Jasmineae, or a ghost lineage that was sister to the Jasmineae. We also created a species tree using only orthogroups with one gene from each species (Fig. S13b). This tree showed the same Oleaceae relationships as the OrthoFinder species tree that included orthogroups containing paralogs as well. With evidence of admixture within the Oleaceae, a network approach, such as what was carried out in Dong et al.[33], would be a more accurate representation of the relationships within the Oleaceae than the phylogeny produced by OrthoFinder, but it can still be utilized for examining polyploidy events. The OrthoFinder gene duplication estimates at nodes (Fig. 3a) agree with the known polyploidy events, showing large increases in well-supported gene duplication events at the known and proposed hexaploidy and WGD events.

To further support the positioning of the polyploidy events within the Oleaceae, we generated a ksrates[59] plot that positioned species splits relative to polyploidy events. Modal distributions of synonymous substitution rate (Ks) values between paralogous *Fraxinus americana* gene pairs represented the two most recent polyploidy events in the evolutionary history of *Fraxinus*: the hexaploidy event at the base of the Oleaceae and the WGD at the base of the Oleeae. Modal distributions of Ks values between orthologous gene pairs between every pair of species were than grouped accordingly, averaged, and corrected relative to the paralogous Ks peaks from *F. americana*. These modes represent the relative timing of the species splits within the Oleaceae. *Jasminum* and *Forsythia* only include the first hexaploidy event, while *Syringa*, *Olea*, *Osmanthus*, and *Fraxinus* contain both polyploidy events (Fig. 3b).

**Conclusion**
We constructed high quality genome assemblies for three critically endangered ash tree species: *Fraxinus americana* (white ash), *F. nigra* (black ash), and *F. pennsylvanica* (green ash). These assemblies had gene space completeness comparable with the recent chromosome-level *F. pennsylvanica* reference assembly (Table 1). We demonstrate that using a single MinION flow cell can be sufficient for in-depth downstream analyses, such as synteny and ploidy evaluations, paleodemographic analyses, and phylogenomics. For example, our assemblies were sufficient to detect multiple polyploidy events in the evolutionary history of

*Fraxinus*. Furthermore, this study was able to identify a six-to-one syntenic gene pair ratio between *Fraxinus* (as well as other published members of the Oleeae) against grapevine (Fig. 1). We also confirm that all sequenced members of the Oleaceae have undergone a hexaploidy event, corroborating the findings of Xu et al.[9] and Wang et al.[16], and that this event was followed by a subsequent WGD in the Oleeae tribe. Our study also showed that use of ONT long reads alone can permit detection of differences in population structure through PSMC analyses, even when compared with PSMC generated from short-read data. We show evidence through PSMC analyses that our *F. nigra* individual likely originates from a more inbred population than our other *Fraxinus* samples (Fig. S10), and correlate that finding with low levels of heterozygosity in the sample (Fig. S2c). With further improvements in ONT error rates, long-read sequencing may have more utility in population structure analyses in the near future.

**Methods**
**Material collection**. Plant material and vouchers were collected from mature trees of unknown sex. *Fraxinus pennsylvanica* vouchers and material for DNA extraction were collected from Point Gratiot Park in Dunkirk, New York. *F. americana* and *F. nigra* vouchers and material for DNA extraction were collected from the College Lodge Forest in Brocton, New York (Table S8). Species identification was done in the field by coauthor Erik Danielson, Stewardship Coordinator with the Western New York Land Conservancy (https://www.wnylc.org/). The collections for material to extract DNA from took place on May 30, 2021, and vouchers from those same trees were collected on July 10, 2021 and are currently stored at the University at Buffalo. Voucher identifications were corroborated in the lab using Gleason and Cronquist[60] and Atha and Boom[61]. From each sample, about 10 g of leaf tissue was placed into separate vials, flash frozen using liquid nitrogen, and stored at −80 °C for later use.

**DNA extraction and sequencing**. We followed the BioNano NIBuffer nuclei isolation protocol and the library preparation procedure from Pacific Biosciences – "Preparing Arabidopsis Genomic DNA for Size-Selected ~20 kb SMRTbellTM Libraries". Additionally, Circulomics Short Read Eliminator (SRE) was used to remove reads less than 25 kb in length with all samples. DNA sequencing was carried out on an ONT' GridION instrument utilizing MinION flowcells (version R9.4). Genomic DNA libraries were prepared with the ONT ligation kit SQK-LSK110. Each flow cell underwent two washes in order to maximize the sequencing output. Sequenced reads were basecalled with the high-accuracy model in Guppy version 5.0.11. Read quality was assessed with Nanostat version 1.5.0 and NanoPlot version 1.38.0[62] (Table S1, Fig. S1).

**Genome assembly**. Flye[63] was utilized for de novo assemblies for the three *Fraxinus* genomes. Multiple Flye versions and assembly options were used to produce the most contiguous and sequence complete assemblies. Flye version 2.8.3 was used for *F. americana* and *F. nigra* and Flye version 2.9 was used for *F. pennsylvanica*. All genomes were assembled using a minimum overlap between reads of 10,000 bp and the scaffolding option. The *F. nigra* assembly benefitted most from three of Flye's internal polishing iterations, while those of *F. americana* and *F. pennsylvanica* benefited the most from one polishing iteration each. Mean genome size estimates were calculated using data from Whittemore et al.[24] Assembly stats were generated using Quality Assessment Tool for Genome Assemblies (QUAST) version 5.0.2[64], and assembly completeness was measured using

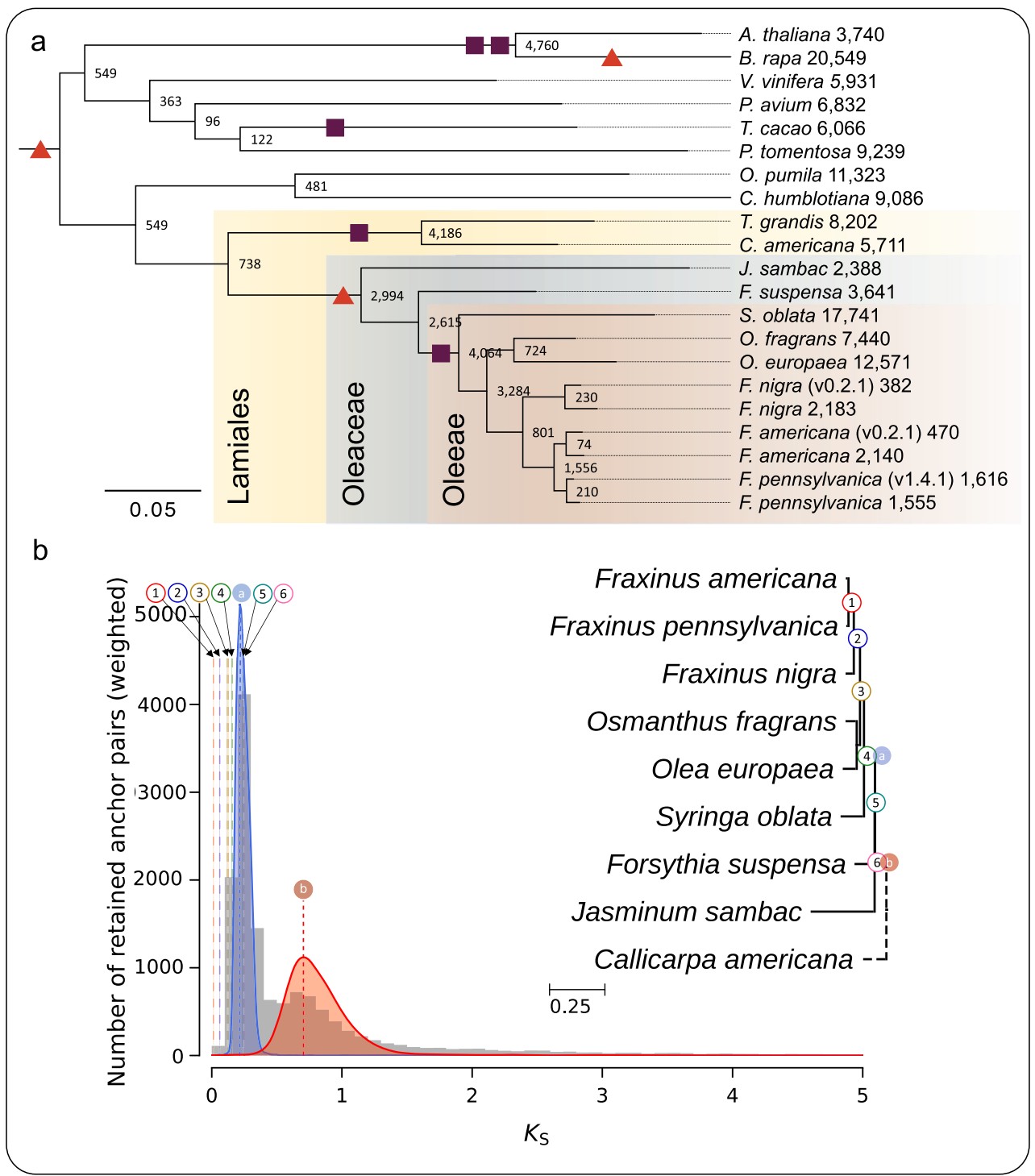

Benchmarking Universal Single-Copy Orthologs (BUSCO) version 5.4.4 embryophyta_odb10[25] (Table S2).

**Genome annotation**. Initial annotations for the *Fraxinus* genome assemblies were generated using Gene Model Mapper (GeMoMa) version 1.9[27,28], a homology-based gene prediction software. Each GeMoMa annotation used the *F. pennsylvanica* reference assembly annotation (Fpennsylvanica_v1.4_genes.gff)[17], the "Full Annotation" (Fraxinus_excelsior_38873_TGAC_v2.gff3) from *Fraxinus excelsior* BATG0.5 assembly[18], *Olea europaea* (GWHAOPM00000000.gff)[14], *Jasminum sambac* (Js.gff)[9], and the *Arabidopsis thaliana* TAIR10 annotation that contains all genes (TAIR10_GFF3_genes.gff)[29] for the reference set. Annotation completeness was measured using BUSCO version 5.4.4 embryophyta_odb10[25] on the predicted proteins file output by GeMoMa (Table 1). BUSCO scores for GeMoMa annotations were corrected using GeMoMa's BUSCORecomputer. Proteins were extracted from the annotation using AGAT version 1.0.0[65], and BUSCO scores were recomputed using Evidential Gene (http://arthropods.eugenes.org/EvidentialGene/).

De novo transposable element (TE) family identification and modeling was completed using RepeatModeler version 2.0.1[66] and RepeatMasker version 4.0.1[67] (Table S3). RepeatModeler Builddatabase and RepeatMasker both used the rmblast search

**Fig. 3 Polyploidy events for Oleaceae and relatives. a** OrthoFinder species tree with inferred gene duplications and associated polyploidy events. Values at nodes and terminal branches are the numbers of gene duplication events with at least 50% of the descendant species maintaining both copies of the duplicated genes. Branch lengths represent average substitutions per site among orthogroup input trees. Red-orange triangles represent known or proposed whole genome "triplication" (hexaploidy) events, with the triangle shared by all members of the tree being the ancient *gamma* hexaploidy event. Purple squares represent known or proposed whole genome duplication (WGD) events. The positioning of hexaploidy and WGD events along branches is arbitrary and does not represent the relative timing of those events. The Lamiales order, Oleaceae family, and Oleeae tribe are highlighted and labeled. Branch support values are displayed in Fig. S13a; these are derived from the proportion of species trees derived from single-locus gene trees supporting each bipartition. **b** Positioning of Oleaceae species splits relative to polyploidy events in *Fraxinus americana*. Syntenic (anchor) Ks paralog pair distribution for *F. americana* is shown in gray. Lognormal mixture model clustering of the median Ks values of the syntenic paralog pairs indicate two putative polyploidy events represented with blue and red curves. Vertical dashed lines labeled a and b denote the modes of these curves and represent the timing of these events. Vertical lines labeled with 1–6 are rate-adjusted modal estimates of one-to-one ortholog Ks distributions between *F. americana* and other species, representing speciation events. Phylogenetic tree branch lengths represent Ks distances estimated from ortholog Ks distributions, with short branches unlabeled. Dashed branches have lengths that cannot be computed by the software. Manually superimposed labels match with the Ks plot and represent *F. americana* polyploidy events and divergence with other lineages on the tree. OrthoFinder tip labels are as follows: (Lamiales, Oleaceae) *F. pennsylvanica*: *Fraxinus pennsylvanica*; *F. pennsylvanica* (v1.4.1): *F. pennsylvanica* (Huff et al.[17]); *F. americana*: *Fraxinus americana*; *F. americana* (v0.2.1): *Fraxinus americana* (Huff et al.[17]); *F. nigra*: *Fraxinus nigra*; *F. nigra* (v0.2.1): *Fraxinus nigra* (Huff et al.[17]); *O. fragrans*: *Osmanthus fragrans*; *O. europaea*: *Olea europaea*; *S. oblata*: *Syringia oblata*; *F. suspensa*: *Forsythia suspensa*; *J. sambac*: *Jasminum sambac*; (other Lamiales) *C. americana*: *Callicarpa americana*; *T. grandis*: *Tectona grandis*; (Gentianales) *C. humblotiana*: *Coffea humblotiana*; *O. pumila*: *Ophiorrhiza pumila*; (Rosids) *T. cacao*: *Theobroma cacao*; *P. tomentosa*: *Populus tomentosa*; *P. avium*: *Prunus avium*; *B. rapa*: *Brassica rapa*; *A. thaliana*: *Arabidopsis thaliana*; *V. vinifera*: *Vitis vinifera*.

engine (http://www.repeatmasker.org/RMBlast.html). In addition to the default RepeatScout[68]/RECON[69] pipeline, the LTR structural discovery pipeline was run as well for RepeatModeler.

**Syntenic mapping, Ks plots, fractionation bias, and pseudo-chromosome assemblies.** Each genome assembly and annotation was uploaded to the Comparative Genomics (CoGe) online platform[70]. Each assembly was input into SynMap2[71] and run against itself to identify polyploidy events. Syntenic dotplots using SynMap's Syntenic Path Assembly option[72] and histograms of the synonymous substitution rates (Ks) between syntenic CDS pairs were generated for each self-vs-self comparison (Figs. S3, S4a-f). Self-self plots were also created for *Osmanthus fragrans* (v1.1 id66037) and *Olea europaea* (v1.0 id63569) (Fig. S5). Each *Fraxinus* species was also run against *Vitis vinifera* (v12x, release 51 id62513; Fig. S14) and the *F. pennsylvanica* reference assembly (v1.4.1 id66054) to ascertain ploidy levels (Fig. S15).

The haploid *Fraxinus* assemblies were scaffolded into pseudo-chromosome-level assemblies using RagTag version 2.1.0[19] and the *F. pennsylvanica* reference assembly[17]. These scaffolded assemblies were annotated using GeMoMa as described above and reuploaded to CoGe. Duplicate subgenomes were assessed to determine relative ploidy level using SynMap's fractionation bias (FractBias) tool[45,46]. Assembly and annotation statistics can be found in supplementary Table S7. Each RagTag assembly was compared against *Vitis vinifera* (v12x, release 51 id62513)[73] (Fig. 1b, c, d) and *Jasminum sambac* (v1.0 id62203)[9] (Fig. S9).

MCScan[74] was utilized to generate syntenic dot plots, syntenic depth, and macrosynteny karyotype plots between assemblies (RagTag assemblies for our *Fraxinus spp.*). Input annotation files were converted to gff3 format using AGAT version 1.0.0[65]. AGAT was also used to extract CDS fasta files using each species' assembly and reformatted annotation. When detecting synteny between two species with the same ploidy level, a C-score cutoff of 0.99 (−cscore = 0.99) was used to filter out older duplication events for clearer connections between syntenic blocks. Otherwise, default options were used to generate figures. GeMoMa[27,28] annotations were also generated for *Forsythia suspensa*[12] and *Osmanthus fragrans*[10] using the *Fraxinus pennsylvanica* reference, *Fraxinus excelsior*, *Olea europaea*, *Jasminum sambac*, and *Arabidopsis thaliana* as references because they lacked publicly available annotations. These new annotations were designated version 1.1.

**Measuring assembly diploidy.** Diploidy of the initial *Fraxinus* assemblies was measured using HapPy (https://github.com/AntoineHo/HapPy). Each *Fraxinus* assembly was indexed using SAMtools version 1.14[75]. ONT raw reads for each sample were mapped onto its corresponding assembly using Minimap2 version 2.20[76]. Aside from not outputting secondary alignments, default options were used. The resulting BAM files were sorted and indexed using SAMtools. Each BAM file was used as input for the HapPy coverage command. Plots were generated using the HapPy estimate command. For *F. americana*, the lower threshold of coverage was 0, the upper threshold of coverage was 35, and the limit for the region between haploid and diploid peaks was 14 (Fig. S2a). For *F. nigra*, the lower threshold of coverage was 0, the upper threshold of coverage was 23, and the limit for the region between haploid and diploid peaks was 1 (Fig. S2c). For *F. pennsylvanica*, the lower threshold of coverage was 0, the upper threshold of coverage was 35, and the limit for the region between haploid and diploid peaks was 16 (Fig. S2e). Haploidy was estimated to 75.78%, 99.57%, and 66.04% for *F. americana*, *F. nigra*, and *F. pennsylvanica*, respectively (Table S4).

**Purging heterozygosity.** The partially diploid assemblies were reduced to haploid assemblies using Purge Haplotigs version 1.1.1[31]. Each assembly was indexed using SAMtools version 1.14[75]. The associated ONT reads were mapped to each assembly using Minimap2 version 2.20[76]. The resulting SAM file was sorted, converted to a BAM file, and indexed using SAMtools[75]. Each indexed and sorted BAM file was used as an input for Purge Haplotigs. The outputs of repetitive elements and their positions in the genome assembly were converted into a BED-format file for input in Purge Haplotigs so those sequences would be ignored during the analysis.

Coverage histograms for each partially diploid assembly were created using Purge Haplotigs (Fig. S16). Based on these plots, the low points between the haploid and diploid peaks were at read depths of 16, 2, and 17 for *F. americana*, *F. nigra*, and *F. pennsylvanica*, respectively. Diploid contigs were purged and genome stats were recalculated using QUAST[64] and BUSCO embryophyta_odb10 [25] (Table 1). HapPy was also run on the haploid assemblies and single peaks were observed for each *Fraxinus* species (Fig. S2b, d, f).

**Population size history**. The reads for each assembly were mapped onto their corresponding assembly using Minimap2 version 2.20[76]. Minimap2 used the ONT preset and did not output secondary alignments. The SAM file was sorted, converted to a BAM file, and indexed using SAMtools version 1.14[75]. SNPs were obtained using bcftools version 1.14[77] mpileup with the ONT configuration and bcftools call using the multiallelic and rare-variant calling model and "-P0.01", outputting variant sites only. An R[78] port for the PSMC model[48], PSMCR, was used to infer population size history (https://github.com/emmanuelparadis/psmcr). The number of PSMCR iterations was set to 30, the largest possible value for time to the most recent common ancestor (maxt) was set to 10, the atomic time interval pattern was "4 + 25*2 + 4 + 6", and 100 bootstrap replicates were calculated. A generation time ($g$) of 15 years and a mutation rate of $7.77 \times 10^{-9}$ substitutions per site per generation were chosen to match previous PSMC parameters used *Prunis persica*[50].

PSMC curves of closely related populations or species may differ in coalescence attributes because of inbreeding (i.e., heterozygosity) differences among accessions[51,79]. Initial PSMC output provides the capacity to estimate pi ($\pi$; nucleotide diversity), where $\pi = 4N_e\mu$. From this formulation, nucleotide diversity, effective population size, and mutation rate are linearly related. From initial PSMC, *F. americana*, *F. nigra*, and *F. pennsylvanica* had 3,458,091, 2,143,896, and 3,928,012 segregating sites, respectively (these are the "sum_n" outputs in the .PSMC files, which we take to reflect heterozygosity). Since $\pi$ roughly equals (number of segregating sites)/(genome size in basepairs), $\pi_{F. nigra} = 0.002763$ and $\pi_{F. pennsylvanica} = 0.004670$, etc., respectively. One can calculate a $\pi_{F. nigra}/\pi_{F. pennsylvanica}$ ratio to describe this difference, i.e., 0.591696, etc. We can then re-run PSMC using adjusted per-generation mutation rates to account for differences in heterozygosites. Of course, $g$ could also be altered (as in Hu et al.[51]) from 15 years to reach a similar end. Using the $\pi_{F. nigra}/\pi_{F. pennsylvanica}$ ratios to adjust $\mu$ to 4.60E-09 for *F. nigra* (etc.) aligns the curves – on the x-axis – to *F. pennsylvanica* at 7.77E-09, in years. Calculations can be found in Supplementary Data 1.

**Phylogenetic orthology inference**. OrthoFinder version 2.5.4[80,81] was used to extract orthologous sequences from a group of Oleaceae, including our *F. americana*, *F. nigra*, and *F. pennsylvanica* (RagTag assemblies), as well as *F. americana* (v0.2.1), *F. nigra* (v0.2.1), and *F. pennsylvanica* (v1.4.1) from Huff et al.[17], *Osmanthus fragrans* (v1.1)[10], *Olea europaea*[14], *Syringa oblata*[16], *Jasminum sambac*[9], and *Forsythia suspensa* (v1.1)[12]. Additional Lamiales (*Callicarpa americana*[82], and *Tectona grandis*[83]), other asterids (*Coffea humblotiana*[84] and *Ophiorrhiza pumila*[85]), and rosids (*Vitis vinifera*[73], *Arabidopsis thaliana*[29], *Brassica rapa*[86], *Theobroma cacao*[87], *Prunus avium*[88], and *Populus tomentosa*[89]) were included. In order to insure only the primary or longest transcripts were included from the *F. americana*, *F. nigra*, and *F. pennsylvanica* assemblies from Huff et al.[17], new CDS files were extracted from these assemblies. CDS and proteins could not be successfully extracted with the publicly available versions using AGAT, so new annotations were generated using GeMoMa. As with all other assemblies generated in this study, annotations were made using the *Fraxinus pennsylvanica* reference, *Fraxinus excelsior*, *Olea europaea*, *Jasminum sambac*, and *Arabidopsis thaliana* annotations as references. These newly annotated versions were denoted by adding a ".1" onto the end of the current version number. Each annotation was reduced to the longest isoform, and proteins were extracted using AGAT version 0.9.1[65]. OrthoFinder was run with default settings. The species tree inference was performed with STAG[90], which uses the proportion of species trees derived from single-locus gene trees supporting each bipartition as its measure of support, and rooted using STRIDE[91]. Lastly, single-copy (orthologs only) gene trees were extracted as input for ASTRAL (version 5.7.4) species tree reconstruction[92]. In total, 200 trees were used as input for ASTRAL and default options were used.

Ksrates version 1.1.3[59] was used to position species splits relative to polyploidy events. Coding sequence (CDS) fasta files were extracted using AGAT version 1.0.0[65]. Paralogous Ks peaks were generated using the RagTag assembly for *Fraxinus americana*. Orthologous Ks peaks were generated using all combinations of species pairs for *F. americana* (v1RagRag), *F. nigra* (v1RagRag), *F. pennsylvanica* (v1RagTag), *Osmanthusfragrans* (v1.1)[10], *Olea europaea*[14], *Syringa oblata*[16], *Jasminum sambac*[9], *Forsythia suspensa* (v1.1)[12], and *Callicarpa americana*[82]. For the input tree, we placed *Forsythia suspensa* sister to the members of the Oleeae and *Jasminum sambac* sister to F. suspensa and the Oleeae to match the phylogenies generated in this study (Figs. 3a, S13). The adjusted Ks values representing divergences (Ks) from *F. americana* were as follows: *F. pennsylvanica* from 0.011937 to 0.013871; *F. nigra* from 0.062818 to 0.061584; *O. fragrans* from 0.123519 to 0.126419; *O. europaea* from 0.136086 to 0.118481; *S. oblata* from 0.177503; *F. suspensa* from 0.210982 to 0.250441; *J. sambac* from 0.403413 to 0.23917.

**Reporting summary**. Further information on research design is available in the Nature Portfolio Reporting Summary linked to this article.

## Data availability

Raw reads for all species are available on the European Nucleotide Archive (ENA; https://www.ebi.ac.uk/). *F. americana* is under the study PRJEB47186 and raw reads are sample accessions SAMEA9806828, SAMEA9806917, and SAMEA9806918. *F. nigra* is under the study PRJEB47212 and raw reads are sample accessions SAMEA9816504, SAMEA9816505, and SAMEA9816506. *F. pennsylvanica* is under the study PRJEB47234 and raw reads are sample accessions SAMEA9816501, SAMEA9816502, and SAMEA9816503. All assembly assembly versions generated in this study are available on the Comparative Genomics (CoGe) online platform (https://genomevolution.org/coge/). Initial Flye assemblies: *F. americana* v0.0 id66026, *F. nigra* v0.0 id66025, *F. pennsylvanica* v0.0 id66056; haploid-purged assemblies: *F. americana* v1 id66137, *F. nigra* v1 id66022, *F. pennsylvanica* v1 id66055; RagTag assemblies: *F. americana* v1Ragtag id66030, *F. nigra* v1Ragtag id66053, *F. pennsylvanica* v1Ragtag id66057; Reannotated Huff et al.[17] assemblies: *F. americana* v0.2.1 id66014, *F. nigra* v0.2.1 id66015, *F. pennsylvanica* v1.4.1 id66054; Reannotated *Forsythia suspensa* v1.1 id66036; Reannotated *Osmanthus fragrans* v1.1 id66037. Genome assemblies, annotations, and source data for figures are available on Dryad (https://doi.org/10.5061/dryad.7sqv9s4xh).

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

## Acknowledgements

We thank Dan Fordham at Oxford Nanopore Technologies plc for the invitation to participate in the Org.One project. Additional support was provided by the National Science Foundation through awards DEB-2030871 to V.A.A. and DEB-2139311 to C.L.

## Author contributions

S.J.F. and V.A.A. conceived the study. S.J.F., C.T., F.A.d.S.C., M.R., N.B., T.K., C.L., V.A.A. sequenced the Oxford Nanopore Technology reads for each species. Location scouting and species identification was carried out by E.S.D. in the field. S.J.F. assembled the genomes and carried out all analyses. S.J.F., C.T., F.A.d.S.C., M.R., E.S.D., C.L., V.A.A. collected samples and vouchers in the field. S.J.F. and V.A.A. wrote the first draft of the paper. All authors provided comments on the first draft. All authors approved the final paper version.

## Competing interests

The authors declare no competing interests.
