## [Peer Review File · Communications Biology]

Reviewers' comments:

Reviewer #1 (Remarks to the Author):

This study presents de novo genome assemblies for three native North American species of ash (genus *Fraxinus*), which have been classified as critically endangered according to the IUCN Red List. Draft genome assemblies are already available for all three species from previous studies, but these were based only on short read data, rather than the long-read sequencing technology used here. The assemblies are of sufficient quality to allow annotation of the gene space and additional downstream evolutionary analyses, and the authors suggest that the same sequencing approach could be used to rapidly generate genome assemblies for other critically endangered plant species.

The methods generally appear well done and the results technically sound, to the best of my knowledge, and the genomic resources generated will provide a useful addition to those already available for plants, and the Oleaceae in particular. However, I have outlined some suggestions below which I believe would further strengthen the manuscript. In particular, I think the paper is currently lacking a strong enough argument for the broader relevance and applicability of the approach presented.

Main Comments

1. Study rationale

The manuscript is lacking a strong argument for why it is important to generate genome assemblies for critically endangered species. In the Introduction, the authors state "With populations of threatened and endangered organisms continually declining, it is vital that high quality genomic records of these species are preserved before they are lost forever", but then they do not go on to say anything further about why this is so vital. How does having a high quality genome assembly from a single individual help us? What is the value of this if the species goes extinct anyway? I do believe that this is a useful endeavour, but the case for doing this is not presented strongly enough by the authors.

2. Wider relevance of sequencing approach

The authors suggest that their sequencing and analysis approach "can provide a relatively fast and inexpensive approach to sequence the 5,232 critically endangered plant species currently on the IUCN Red-List". However, the test case here, based on three *Fraxinus* species, could be argued to be a relatively simple scenario. All three species have comparatively small genome sizes and are diploid (albeit with WGDs in their relatively recent evolutionary history), but many plant species are functional polyploids and may have much larger genome sizes. How would this approach cope with cases such as those? It is not explicitly mentioned why the other two critically endangered native North American *Fraxinus* species were not included in the study, but I wonder whether *F. profunda* was excluded because it is a hexaploid with a much larger genome size? I think the authors should make some further comment on issues such as genome size and ploidy and to what extent these might complicate, or not, the use of the approach they advocate. There are publicly accessible datasets on plant C-values and ploidy – could the authors do a quick analysis of the distribution of genome sizes in the 5,232 critically endangered plant species and look at the proportion of recent polyploids to better inform the discussion?

3. Comparison with other large-scale genome sequencing initiatives

The authors describe the Org.ONE project and its aim to sequence and assemble the genomes of critically endangered species. However, as I am sure the authors are already aware, there are other initiatives aiming to sequence, assemble and annotate the genomes for large numbers of species. Foremost amongst these is the Earth BioGenome Project, which aims to sequence all eukaryotic species. I find it strange that there is no mention/discussion of these other efforts. How does the approach they suggest compare with that used by other initiatives? Is it complementary (is the Org.ONE project affiliated with the Earth Biogenome project?), or is there actually a risk that efforts will be duplicated? The authors should address these questions.

Minor Comments

Introduction

4. It would be worth mentioning that the three ash species sequenced in the study come from two separate clades within the genus, with *F. nigra* actually being more closely related to European ash (*F. excelsior*) than to the other native North American species.
5. "For example, *F. pennsylvanica*, another member of section *Melioides*, displays loss of function in one candidate gene due to two frame shift mutations and a loss of function mutation in another candidate gene due to a stop codon gain" – the study cited did not experimentally verify the impact of these mutations on gene function, so they should be referred to as "putative" loss-of-function mutations.
6. "Huff et al. [17] generated new 800 basepair (bp) insert size library data for eight of these assemblies" – it would be more correct to say they generated them for "eight of these individuals".

Results and Discussion

7. In relation to the GeMoMa annotation of the newly generated assemblies, the authors say "The annotation of Huff et al.'s *F. pennsylvanica* reference assembly was not used because it only contained 82.6% complete BUSCOs", but couldn't both the *F. excelsior* reference annotation and the *F. pennsylvanica* annotations have been used in combination? As the annotation from Huff et al., was de novo rather than reference based it may contain additional *Fraxinus* specific genes that are missing from the *F. excelsior* annotation, despite having fewer complete BUSCOs. It would be interesting to see if using both of these annotations together changes the number of genes annotated for the newly generated long read assemblies.
8. "Our resulting *F. americana*, *F. nigra*, and *F. pennsylvanica* annotations contained 37,952, 36,821, and 37,079 gene models, constituting 96.7%, 97.3%, and 96.8% complete BUSCOs, respectively (Table 1)" – I think it would make more sense to say "encompassing 96.7% ...", because "constituting" makes it sound like there are tens of thousands of BUSCOs.
9. "This was corroborated by greater-than-expected gene model prediction numbers and duplicated BUSCO counts compared with the *F. pennsylvanica* reference in these initial assemblies, particularly for *F. americana* and *F. pennsylvanica* (Table S2)" – also, the fact that the initial assemblies for *F. americana* and *F. pennsylvanica* were significantly larger than the estimated 1C genome sizes for these taxa (Table S2) indicates the presence of separately assembled haplotypes.
10. When mapping the ONT reads back to the assemblies, the coverage for *F. nigra* is significantly lower (c. half) than both that based on the raw reads and that for the other species based on read mapping. Could the authors comment on the cause of this?

Materials and Methods

11. "The collections for material to extract DNA from took place on May 30, 2021, and vouchers from those same trees were collected on July 10, 2021" – please indicate where the vouchers have been deposited.
12. "Flye version 2.8.3[50], and for *F. pennsylvanica*, Flye version 2.9 was used" – what is the reason for using different versions of the assembler? Are there significant differences between the versions that could confuse the comparison of the assembly stats across taxa?
13. "Annotation completeness was measured using BUSCO version 4.1.4 *embryophyta_odb10*" – it would be useful if the authors stated, either in the text or as a footnote to the relevant table, the total number of BUSCOs in this database (i.e. how many BUSCOs were searched for in total).
14. "New GeMoMa annotations were generated for the *F. americana*, *F. nigra*, and *F. pennsylvanica* assemblies from Huff et al. because CDS and proteins could not be successfully extracted with the publicly available versions" – I'm a bit confused why the CDS for the existing *F. pennsylvanica* assembly annotation at least could not be used, since there is a file of the CDS on Zenodo (<https://zenodo.org/record/5176117#.Y7gbi-zP1-U>), or were the authors wanting the output from the previous GeMoMa annotation specifically?

Reviewer #2 (Remarks to the Author):

Fleck et al report on the use of ONT to produce assemblies for three threatened tree species: green, white and black ash. They use these assemblies to explore the history of genome polyploidization events and population demographic history, as well as calculate eudicot orthogroups. The paper is clearly written.

Major points:

1. The authors refer to their genomes as “high quality” – however, as the point of the paper seems to be to confirm the org.ONE strategy of ONT assemblies, these are not high quality in terms of today’s standards. Output is thousands to tens of thousands of scaffolds, which are not able to be oriented into chromosomes without a closely related preexisting chromosome-scale genome (in this case), a dense genetic map and/or other scaffolding techniques like Hi-C. This point needs to be made clearly. In particular, *F. nigra* is in a different section of the genus, and there could be major chromosomal rearrangements for black ash that have not been detected and won’t be until there is scaffolding independent of the existing green ash assembly.
2. The org.ONE project is an exciting opportunity for critically endangered species. It features prominently in the abstract and the introduction and not at all in the discussion. The discussion should contextualize this work within the a conservation framework. Why is genome sequencing helpful for critically endangered species in general and ash specifically? The analyses presented here are interesting in terms of genome evolution but not tied to any sort of conservation efforts.
3. The section on fractionation bias needs a better explanation – what is it and what is it telling you? Line 256 “corresponding patterns” – is this referring to panels c and d of Figure S12? There is a somewhat parallel pattern, but fractionation bias usually is opposite, where regions of a chromosome from one subgenome are maintained while homologous regions in the opposite subgenome chromosome are highly diverged. Also, line 258 “indicating that there was indeed a WGD” - Fractionation doesn’t provide evidence of a WGD itself, but it does show gene order preservation/loss assuming the WGD has been correctly determined. If you are keeping this analysis, the figure could be moved to the main text.
4. The paleodemography analysis does not seem to align with what is already known about the evolution of *Fraxinus*. *Fraxinus* evolved in NA and then a migration to Eurasia occurred; *F. nigra* is from a late intercontinental migration from Eurasia back to NA in the Miocene. (This is all worth mentioning in the manuscript). In any case, the last common ancestor of black ash to other NA ash is quite old. Hinsinger 2013 reports on this, putting *F. nigra* coming back over to NA around 15-20 MYA and the last common ancestor way back in the Eocene, 45Mya or so. You assumption that the PSMCR should coalesce around a “same, pre-speciation stem lineage” (Line 274) is far-fetched, as the analysis will likely not include history that far back in time. Is the PSMCR plot x axis “years before present”? This needs to be labeled. If so then the prespeciation lineage is estimated at 50,000-500,000 years? Overall, the justification for altering the mutation rate and the location of the presumed *Fraxinus* stem lineage needs clarification.
5. Line 292 – “As such, our *F. pennsylvanica* individual may come from a population with even more interspecies admixture than the Huff et al. reference accession, which may stem from a population with less admixture than previously thought.” The Huff paper had range wide green ash species representation and did not suggest landscape level interspecies hybridization or introgression. (The admixture was for populations) But yes, green ash does interbreed successfully with white ash, so perhaps you sampled a hybrid or a population with a history of introgression? This would be good background to give to support the suggestion of an “interspecies admixture”.

Minor:

Line 118: "in the evolution the" -> add "in the evolution of the"

CDS acronym not defined

Line 249 - "famiy" -> "family"

Line 279 - "pensylvanica" misspelled

Line 280 - "amerciana" misspelled

Line 286 - "pensylvanica" misspelled

Line 304- "infrspecifically" -> "intraspecifically"? But generally an awkward construction that might benefit from rewording

Line 381 - "benefitted" -> "benefited"

Line 422 - "pensylvanica" misspelled

Line 425 - "pensylvanica" misspelled

Reviewer #3 (Remarks to the Author):

The authors reported the assemblies of the three North American ash tree genomes as well as their genomic evolutionary features. This work is meaningful, which is important for genetic information preservation of endangered species. However, there is a big concern about the quality of the assembly because only ONT reads were used to assemble. First, the assembly seems fragile, and second, they do not show any data to prove the quality of the base. Although Oxford Nanopore technology can provide a relatively fast and inexpensive approach, the qualities of the assembly and the genome sequence have not reached a relatively high and acceptable level. If the sequence quality is low, the following analyses such as gene annotation, k_a/k_s calculation, gene duplication and WGD may not be correct. Therefore, I strongly recommend that the author should use other sequencing technologies such as (PacBio, illumine and HiC reads) to improve the quality of the assembly and sequence. Otherwise, I can not encourage to publish this manuscript.

We thank the reviewers for their constructive comments. Please refer to our description below of all revisions made, interspersed in blue among the referees' comments.

Reviewers' comments:

Reviewer #1 (Remarks to the Author):

This study presents de novo genome assemblies for three native North American species of ash (genus *Fraxinus*), which have been classified as critically endangered according to the IUCN Red List. Draft genome assemblies are already available for all three species from previous studies, but these were based only on short read data, rather than the long-read sequencing technology used here. The assemblies are of sufficient quality to allow annotation of the gene space and additional downstream evolutionary analyses, and the authors suggest that the same sequencing approach could be used to rapidly generate genome assemblies for other critically endangered plant species.

The methods generally appear well done and the results technically sound, to the best of my knowledge, and the genomic resources generated will provide a useful addition to those already available for plants, and the Oleaceae in particular. However, I have outlined some suggestions below which I believe would further strengthen the manuscript. In particular, I think the paper is currently lacking a strong enough argument for the broader relevance and applicability of the approach presented.

Main Comments

1. Study rationale

The manuscript is lacking a strong argument for why it is important to generate genome assemblies for critically endangered species. In the Introduction, the authors state "With populations of threatened and endangered organisms continually declining, it is vital that high quality genomic records of these species are preserved before they are lost forever", but then they do not go on to say anything further about why this is so vital. How does having a high quality genome assembly from a single individual help us? What is the value of this if the species goes extinct anyway? I do believe that this is a useful endeavour, but the case for doing this is not presented strongly enough by the authors.

We thank the reviewer for this comment and agree that it should be addressed in the manuscript. A small section was added to the introduction that covers some ways in which reference genomes aid in conservation management. We also included an example of a reference genome that was not chromosome-level that had impacts in conservation management.

2. Wider relevance of sequencing approach

The authors suggest that their sequencing and analysis approach "can provide a relatively fast and inexpensive approach to sequence the 5,232 critically endangered plant species currently on the IUCN Red-List". However, the test case here, based on

three *Fraxinus* species, could be argued to be a relatively simple scenario. All three species have comparatively small genome sizes and are diploid (albeit with WGDs in their relatively recent evolutionary history), but many plant species are functional polyploids and may have much larger genome sizes. How would this approach cope with cases such as those? It is not explicitly mentioned why the other two critically endangered native North American *Fraxinus* species were not included in the study, but I wonder whether *F. profunda* was excluded because it is a hexaploid with a much larger genome size? I think the authors should make some further comment on issues such as genome size and ploidy and to what extent these might complicate, or not, the use of the approach they advocate. There are publicly accessible datasets on plant C-values and ploidy – could the authors do a quick analysis of the distribution of genome sizes in the 5,232 critically endangered plant species and look at the proportion of recent polyploids to better inform the discussion?

We thank the reviewer for pointing this out and agree that these methods cannot be used to sequence all 5,232 critically endangered plant species currently on the IUCN Red-List due to genome size restrictions. Currently, a single MinION flow cell will only produce a limited depth of coverage and genomes with a 1C value larger than 1 GB may not work. We chose the *Fraxinus* spp. for this study based on the availability to collect locally within the United States and sequence in our own lab, a requirement of the Org.ONE project. Our lab does not focus on *Fraxinus* or Oleaceae at present, but the senior author (V.A. Albert published a highly cited paper on Oleaceae molecular systematics over 20 years ago, so the present work represented a satisfying conclusion of sorts for a longstanding interest in the family. On sequencing other species of *Fraxinus*, if we had sequenced the octoploid *F. profunda*, which has a 1C value of 3.814 pg or 3.730 GB, it would have been unlikely that the sequencing would have reached a high enough depth of coverage for a quality assembly. *F. quadrangulata* would have been an easy enough addition to the study had we been able to identify one locally, which we could not. This species has a 2n value of 46 like the others we sequenced and would likely have had the smallest 1C genome size out of the group at 0.725 pg or 709.1 MB. While scaffolding the draft assemblies using RagTag was only possible because we had access to the *F. pennsylvanica* reference assembly, which happened to have the same chromosome number and ploidy level as our species, we collected material for sequencing before Huff et al. had published their paper.

3. Comparison with other large-scale genome sequencing initiatives

The authors describe the Org.ONE project and its aim to sequence and assemble the genomes of critically endangered species. However, as I am sure the authors are already aware, there are other initiatives aiming to sequence, assemble and annotate the genomes for large numbers of species. Foremost amongst these is the Earth BioGenome Project, which aims to sequence all eukaryotic species. I find it strange that there is no mention/discussion of these other efforts. How does the approach they suggest compare with that used by other initiatives? Is it complementary (is the Org.ONE project affiliated with the Earth Biogenome project?), or is there actually a risk that efforts will be duplicated? The authors should address these questions.

We thank the reviewer for bringing up this point. Org.ONE is not affiliated with the Earth Biogenome project. The goal of Org.ONE is to empower local researchers to sequence and assemble genomes of critically endangered species close to the sample's origin. With projects such as the Earth Biogenome project, samples often need to be shipped out of their country of origin for sequencing. It is possible that some overlap will be accomplished from the two projects. In some cases, two individuals from the same species within different populations will be sequenced, which could in fact wind up useful in some cases. In other cases, two individuals from the same species within the same populations might be sequenced, which may have its own scientific advantages. Regardless, the scopes of Org.ONE and Earth Biogenome are outside of our control, and this paper is not meant as a justification for Org.ONE, merely as a single exemplar case proving its effectiveness for moderately sized plant genomes

Minor Comments

Introduction

4. It would be worth mentioning that the three ash species sequenced in the study come from two separate clades within the genus, with *F. nigra* actually being more closely related to European ash (*F. excelsior*) than to the other native North American species.

We agree that this is important to mention. After revising the manuscript, this point is especially important for the divergence time between *F. nigra* in the *Fraxinus* section and *F. americana* and *F. pennsylvanica* in the *Melioides* section. It's further important for us to point this out because *F. pennsylvanica* was used to scaffold *F. nigra*, despite being moderately distantly related. Potentially, there could be major rearrangements between the two genomes that will not be represented in *F. nigra*'s scaffolded assembly. However, we are confident given the high degree of synteny among all of our genomes that our assemblies represent a useful advance for investigators interested in Oleaceae genomes.

5. "For example, *F. pennsylvanica*, another member of section *Melioides*, displays loss of function in one candidate gene due to two frame shift mutations and a loss of function mutation in another candidate gene due to a stop codon gain" – the study cited did not experimentally verify the impact of these mutations on gene function, so they should be referred to as "putative" loss-of-function mutations.

We agree with this point and updated it in the manuscript.

6. "Huff et al. [17] generated new 800 basepair (bp) insert size library data for eight of these assemblies" – it would be more correct to say they generated them for "eight of these individuals".

We agree with this point and updated it in the manuscript.

Results and Discussion

7. In relation to the GeMoMa annotation of the newly generated assemblies, the authors say "The annotation of Huff et al.'s *F. pennsylvanica* reference assembly was not used

because it only contained 82.6% complete BUSCOs”, but couldn’t both the *F. excelsior* reference annotation and the *F. pennsylvanica* annotations have been used in combination? As the annotation from Huff et al., was de novo rather than reference based it may contain additional *Fraxinus* specific genes that are missing from the *F. excelsior* annotation, despite having fewer complete BUSCOs. It would be interesting to see if using both of these annotations together changes the number of genes annotated for the newly generated long read assemblies.

We thank the reviewer for this suggestion. We had at first considered that one *Fraxinus* reference was sufficient for annotating our genomes with GeMoMa. Early annotations were made with the *F. pennsylvanica* reference (v1.4) or *F. excelsior* (vBATG-0.5), and *F. excelsior* (vBATG-0.5) gave better results. Both had de novo genome annotations, so we prioritized *F. excelsior* over the *F. pennsylvanica* reference. It is true that we could have used both species as references for GeMoMa in the manuscript. To see if there was a difference in the number of BUSCOs or gene models, we reran GeMoMa with the *F. pennsylvanica* reference assembly included alongside *Fraxinus excelsior*, *Olea europaea*, *Jasminum sambac*, and *Arabidopsis thaliana* as references. With *F. pennsylvanica* included, there were 3,318 more gene models predicted and four fewer missing BUSCOs for *F. americana*. *F. nigra* had 838 more gene models and five fewer missing BUSCOs. *F. pennsylvanica* had 3522 more gene models and seven fewer missing BUSCOs. The reviewer was correct that many more genes were found by including both de novo *F. excelsior* and *F. pennsylvanica* annotations as GeMoMa references. With this new knowledge, we regenerated all annotations to include the *F. pennsylvanica* reference and redid all analyses with the new, improved annotations.

8. “Our resulting *F. americana*, *F. nigra*, and *F. pennsylvanica* annotations contained 37,952, 36,821, and 37,079 gene models, constituting 96.7%, 97.3%, and 96.8% complete BUSCOs, respectively (Table 1)” – I think it would make more sense to say “encompassing 96.7% ...”, because “constituting” makes it sounds like there are tens of thousands of BUSCOs.

We agree with this point and have updated the manuscript.

9. “This was corroborated by greater-than-expected gene model prediction numbers and duplicated BUSCO counts compared with the *F. pennsylvanica* reference in these initial assemblies, particularly for *F. americana* and *F. pennsylvanica* (Table S2)” - also, the fact that the initial assemblies for *F. americana* and *F. pennsylvanica* were significantly larger than the estimated 1C genome sizes for these taxa (Table S2) indicates the presence of separately assembled haplotypes.

We agree with this point and have updated the manuscript to mention this specifically.

10. When mapping the ONT reads back to the assemblies, the coverage for *F. nigra* is significantly lower (c. half) than both that based on the raw reads and that for the other species based on read mapping. Could the authors comment on the cause of this?

We thank the reviewer for pointing this out. When mapping long-reads onto each initial Flye assembly, *F. americana*, *F. nigra*, and *F. pennsylvanica* had an average of 18.4211, 21.1345, and 17.0895x coverage, respectively. When HapPy plotted the coverage, all three individuals had peaks around 11x and *F. americana* and *F. pennsylvanica* had a secondary peak at 20x. When mapping long-reads onto each haploid assembly, *F. americana*, *F. nigra*, and *F. pennsylvanica* had an average of 22.7905x, 21.1998x, and 21.1619x coverage, respectively. When HapPy plotted the coverage, *F. nigra* still had a single peak at 11x and *F. americana* and *F. pennsylvanica* have a single peak at 20x.

Materials and Methods

11. “The collections for material to extract DNA from took place on May 30, 2021, and vouchers from those same trees were collected on July 10, 2021” – please indicate where the vouchers have been deposited.

We agree that this should be added and updated this information in the manuscript.

2. “Flye version 2.8.3[50], and for *F. pennsylvanica*, Flye version 2.9 was used” - what is the reason for using different versions of the assembler? Are there significant differences between the versions that could confuse the comparison of the assembly stats across taxa?

For all three species, we tried different assembly options for Flye version 2.8.3 and 2.9. This was because Flye version 2.9 was released right around the same time we were generating our genome assemblies and we wanted to see if the new version made improvements. The new version only improved the *F. pennsylvanica* assembly.

13. “Annotation completeness was measured using BUSCO version 4.1.4 embryophyta_odb10 “ – it would be useful if the authors stated, either in the text or as a footnote to the relevant table, the total number of BUSCOs in this database (i.e. how many BUSCOs were searched for in total).

We agree with this point and updated the manuscript accordingly. “Total BUSCOs searched” was added to every table that reported BUSCOs.

14. “New GeMoMa annotations were generated for the *F. americana*, *F. nigra*, and *F. pennsylvanica* assemblies from Huff et al. because CDS and proteins could not be successfully extracted with the publicly available versions “ – I’m a bit confused why the CDS for the existing *F. pennsylvanica* assembly annotation at least could not be used, since there is a file of the CDS on Zenodo (<https://zenodo.org/record/5176117#.Y7gbi-zP1-U>), or were the authors wanting the output from the previous GeMoMa annotation specifically?

There were a few reasons we reannotated these genomes. First, we wanted to make sure there weren't any alternate splice forms included in the protein fasta file. Typically, this is easy to check by comparing the number of “mRNA” features in the gff file

matching the number of “gene” features. The reference guided assemblies from Huff et al. used an annotation that was output by gFACs, which outputs an Ensembl v3 gff3 that only contains information on mRNA, exon, and intron types. This meant that comparing gene and mRNA numbers would be complicated. We were also having difficulties using AGAT to reduce the gff file to just the longest transcript and then re-extract the proteins for OrthoFinder. Additionally, we wanted to upload these genomes and annotations to <https://genomeevolution.org/coge/>, which requires that CDS features are defined in the gff file, and we didn’t have a simple way of adding them in. Lastly, when we ran BUSCO on the publicly available protein files, that approach found fewer than 90% complete embryophyta_odb10 BUSCOs, while our annotations found greater than 95% complete embryophyta_odb10 BUSCOs. With each issue in mind, we decided to use new annotations that were of a consistent quality in order to get a better idea of the differences between the genome assemblies themselves.

Reviewer #2 (Remarks to the Author):

Fleck et al report on the use of ONT to produce assemblies for three threatened tree species: green, white and black ash. They use these assemblies to explore the history of genome polyploidization events and population demographic history, as well as calculate eudicot orthogroups. The paper is clearly written.

Major points:

1. The authors refer to their genomes as “high quality” – however, as the point of the paper seems to be to confirm the org.ONE strategy of ONT assemblies, these are not high quality in terms of today’s standards. Output is thousands to tens of thousands of scaffolds, which are not able to be oriented into chromosomes without a closely related preexisting chromosome-scale genome (in this case), a dense genetic map and/or other scaffolding techniques like Hi-C. This point needs to be made clearly. In particular, *F. nigra* is in a different section of the genus, and there could be major chromosomal rearrangements for black ash that have not been detected and won’t be until there is scaffolding independent of the existing green ash assembly.

We fully understand this point and agree that we needed to be more clear about the ways in which our assemblies can be considered “high quality”. This has been addressed in the manuscript. For example, our statement is in part supported by the fact that while our genomes are not chromosome-level, they’re much more contiguous than the short-read assemblies used by Huff et al. (2022) for reference-guided scaffolding using RagTag. Furthermore, RagTag was able to scaffold more of our long-read assemblies compared to the short read assemblies. For Huff et al, only 78.25% and 66.44% of the short-read *F. americana* and *F. nigra* assemblies, respectively, were scaffolded. For this study, 95.40%, 97.29%, and 95.74% of our long-read *Fraxinus americana*, *F. nigra*, and *F. pennsylvanica* assemblies, respectively, were scaffolded. Additionally, our long-read assemblies have high gene space accuracy as shown by the BUSCO scores, with all haploid assemblies having greater than 95% complete embryophyta_odb10 BUSCOs. They’re also reliably haploid with 1C assembly sizes closer to the flow cytometry estimations from Whittmore et al. (2018) than the already published short-read and reference-guided assemblies.

We agree with the concern about scaffolding *Fraxinus nigra* with *F. pennsylvanica* and have addressed this in the manuscript.

2. The org.ONE project is an exciting opportunity for critically endangered species. It features prominently in the abstract and the introduction and not at all in the discussion. The discussion should contextualize this work within the a conservation framework. Why is genome sequencing helpful for critically endangered species in general and ash specifically? The analyses presented here are interesting in terms of genome evolution but not tied to any sort of conservation efforts.

We thank the reviewer for this comment and agree that it should be addressed in the manuscript. A small section was added to the introduction that covers some ways in which reference genomes aid in conservation management. While this project was sequenced with the aid of Org.ONE, which is focused on making sequencing data for critically endangered species publicly available, we focused on demonstrating the genome quality that could be generated from a single MinION flow cell. Reference genome quality is also vital for conservation genetics, and we include a brief example of the Tasmanian devil (*Sarcophilus harrisi*), which had a similarly contiguous reference genome (35,974 scaffolds, N50 1.85 Mb) that had major impacts in conservation management of that species.

3. The section on fractionation bias needs a better explanation – what is it and what is it telling you? Line 256 “corresponding patterns” – is this referring to panels c and d of Figure S12? There is a somewhat parallel pattern, but fractionation bias usually is opposite, where regions of a chromosome from one sub-genome are maintained while homologous regions in the opposite sub-genome chromosome are highly diverged. Also, line 258 “indicating that there was indeed a WGD “ - Fractionation doesn’t provide evidence of a WGD itself, but it does show gene order preservation/loss assuming the WGD has been correctly determined. If you are keeping this analysis, the figure could be moved to the main text.

We agree that fractionation bias needs to be explained more clearly. This has been addressed in the section explaining Figure 1b,c,d and S10b,c. Both *Fraxinus* and *Jasminum* against *Vitis* display the same pattern of 2/3 of their sub-genomes having similar patterns of high gene retention and 1/3 of their sub-genomes having a low level of gene retention. These patterns suggest an allopolyploid triplication event in the ancestor of *Jasminum* and *Fraxinus* since their divergence from *Vitis*. This may also indicate sub-genome dominance, but we did not look at gene expression in this project to confirm that is indeed happening in *Fraxinus*. This was also briefly addressed in the section explaining S12b,c,d. When fractionation bias is run between *Fraxinus* and *Jasminum*, there are two *Fraxinus* sub-genomes with similar patterns of gene retention against a single *Jasminum* chromosome with no clear biased gene retention. This may indicate an autopolyploid duplication event in the ancestor of *Fraxinus* (and the rest of the Oleaceae) since it’s divergence with *Jasminum*.

4. The paleodemography analysis does not seem to align with what is already known about the evolution of *Fraxinus*. *Fraxinus* evolved in NA and then a migration to Eurasia occurred; *F. nigra* is from a late intercontinental migration from Eurasia back to NA in the Miocene. (This is all worth mentioning in the manuscript). In any case, the last common ancestor of black ash to other NA ash is quite old. Hingsinger 2013 reports on this, putting *F. nigra* coming back over to NA around 15-20 MYA and the last common ancestor way back in the Eocene, 45Mya or so. Your assumption that the PSMCR should coalesce around a “same, pre-speciation stem lineage” (Line 274) is far-fetched, as the analysis will likely not include history that far back in time. Is the PSMCR plot x axis “years before present”? This needs to be labeled. If so then the prespeciation lineage is estimated at 50,000-500,000 years? Overall, the justification for altering the mutation rate and the location of the presumed *Fraxinus* stem lineage needs clarification.

We thank the reviewer for bringing this to our attention and this section of the manuscript has been reworked. The speciation events between our *Fraxinus* species are likely too old for PSMC to detect. We also address the dating of these speciation events. While Hingsinger et al. (2013) dated the split between the *Fraxinus* section and the *Meloides* section to be about 45 mya, many subsequent studies would place that event much closer to present day (i.e. Unver et al. (2017), Dong et al. (2022), Qi et al. (2022)). We still show the similarity in their demographic histories without claiming a coalescence of their populations. The focus has been shifted more to comparing the PSMC generated from the short-read *F. pennsylvanica* reference data with the PSMC for our *F. pennsylvanica* generated from long-read data (Figure 2b). It's likely that these populations coalesce within the last million years. We also address some issues with estimating mutation rate and generation time for *Fraxinus* and settle on a mutation rate measured from Xie et al. (2020) for peach (*Prunus persica*) and a generation time of fifteen years. We failed to find these values specifically measured for *Fraxinus* and need to apply some caution with believing the dates for specific demographic events that we observe in the PSMC. In terms of justifying altering the mutation rate to correct for more inbred populations, we stand by the position that the simulated selfing data provided strong evidence for this in Hu et al.'s (2022) lychee paper. They demonstrated that Inbred populations have PSMC plots that are shifted toward smaller population sizes and more recent times (Supplementary Note Fig. 2). We also make sure to note that the population that the published *F. pennsylvanica* reference comes from may not be more inbred than the population we sampled our *F. pennsylvanica* from. There may also be differences with long-read data, such as an increased error rate, that artificially shifts the demographic curve toward larger effective population sizes and older times.

5. Line 292 – “As such, our *F. pennsylvanica* individual may come from a population with even more interspecies admixture than the Huff et al. reference accession, which may stem from a population with less admixture than previously thought.” The Huff paper had range wide green ash species representation and did not suggest landscape level interspecies hybridization or introgression. (The admixture was for populations) But yes, green ash does interbreed successfully with white ash, so perhaps you

sampled a hybrid or a population with a history of introgression? This would be good background to give to support the suggestion of an “interspecies admixture”.

We agree with this point, and it has been addressed in the manuscript.

Minor:

Line 118: “in the evolution the” → add “in the evolution of the”

CDS acronym not defined

Line 249 – “famiy” → “family”

Line 279 – “pensylvanica” misspelled

Line 280 – “amerciana” misspelled

Line 286 – “pensylvanica” misspelled

Line 304- “infrspecifically” → “intraspecifically”? But generally an awkward construction that might benefit from rewording

Line 381 – “benefitted” → “benefited”

Line 422 – “pensylvanica” misspelled

Line 425 – “pensylvanica” misspelled

We agree with all of the minor issues, and they have all been fixed in the manuscript.

Reviewer #3 (Remarks to the Author):

The authors reported the assemblies of the three North American ash tree genomes as well as their genomic evolutionary features. This work is meaningful, which is important for genetic information preservation of endangered species. However, there is a big concern about the quality of the assembly because only ONT reads were used to assemble. First, the assembly seems fragile, and second, they do not show any data to prove the quality of the base. Although Oxford Nanopore technology can provide a relatively fast and inexpensive approach, the qualities of the assembly and the genome sequence have not reached a relatively high and acceptable level. If the sequence quality is low, the following analyses such as gene annotation, ka/ks calculation, gene duplication and WGD may not be correct. Therefore, I strongly recommend that the author should use other sequencing technologies such (Pacbio, illumine and HiC reads) to approve the quality of the assembly and sequence. Otherwise, I can not encourage to publish this manuscript.

We thank the reviewer for their comment, but disagree that the basic quality of these assemblies is questionable. While it is typical to correct reads with Illumina short-read data, without correcting the reads, over 95% complete embryophyta_odb10 BUSCOs were identified in each assembly and annotation (Table 1, S2, S6). Oxford Nanopore data have become significantly more accurate, and with sufficient depth, many remaining errors can be corrected using Nanopore reads only. Furthermore, after we reduced our assemblies down to haploid assemblies, they were all very close to their estimated genome sizes from flow cytometry in Whittmore et al. (2018) (Table 1, S2, S6). Additionally, Figure S4 shows that long-read and short-read assemblies have corresponding Ks peaks, which represent whole genome duplications, in virtually identical locations: One at about $\log_{10} = -0.6$ and one at about $\log_{10} = -0.15$. Additionally,

our Ksrates analysis placed the species splits within the Oleaceae (represented by orthologous Ks peaks between 2 species) relative to the polyploidy events within that family (represented by paralogous Ks peaks in *F. americana*) as we had predicted with other analyses and by other recent publications: *Forsythia* and *Jasminum* only had the hexaploidy event at the base of the Oleaceae, and the rest of the Oleaceae contained a subsequent whole genome duplication (Figure 3b). Furthermore, our long-read assemblies had very strong synteny with published high-quality genome assemblies. These assemblies ranged from distant relatives like *Vitis vinifera* (Figure 1, S9), and close relatives like *Olea europaea*, *Osmanthus fragrans*, *Syringa oblata* (Figure S6), and *Jasminum sambac* (Figure S12). For the synteny analyses, we used assemblies that were scaffolded using Huff et al. (2022)'s *F. pennsylvanica* reference assembly, but the scaffolding process does not correct the sequences of the input assembly or fill in gaps between scaffolded contigs. Homologous and syntenic gene pairs were identified using only sequence matches from the long-read assemblies. Additionally, when included in an OrthoFinder analyses, each long-read assembly paired together with their short-read counterpart in the species tree output (Figure 3a, S16). Lastly, when we tried scaffolding our assemblies with the *F. pennsylvanica* reference assembly using RagTag, much more of our assemblies were scaffolded into pseudo-chromosomes. For Huff et al, only 78.25% and 66.44% of the short-read *F. americana* and *F. nigra* assemblies, respectively, were scaffolded. For this study, 95.40%, 97.29%, and 95.74% of our long-read *Fraxinus americana*, *F. nigra*, and *F. pennsylvanica* assemblies, respectively, were scaffolded. All of these results taken together suggest that the long-read assemblies should be as trustworthy as, and arguably have more utility than, the short read and reference-guided assemblies that have already been published.

Reviewers' comments:

Reviewer #1 (Remarks to the Author):

This is a revised version of a manuscript that I reviewed previously. The authors have made detailed responses to the points raised by all three reviewers and I am satisfied overall with the changes they have made to the text and analyses. There is however one point that I raised previously that I feel still needs further clarification – this is why the coverage for *F. nigra* in some of the analyses is estimated at about half that expected. I also have a few additional minor comments, which are outlined further below.

1. In response to my previous comment on this topic (number 10 in my original review), the authors outline how on the basis of mapping long reads either to the initial assembly for *F. nigra*, or to the haploid assembly, the estimated coverage is c. 21x but that both before and after purging of haplotigs the estimated coverage with HapPy is 11x. However, this does not explain why this is the case. If this species truly had very few haplotigs in the initial assembly and the coverage of the genome is truly at around 21x, as suggested from the read mapping, then we would expect to see a peak at c. 21x from the HapPy coverage analysis, not 11x. Also, it makes no sense that the inferred threshold for coverage between the diploid and haploid parts of the assembly is at 1x for an assembly where it is suggested there is actually 21x coverage – surely this threshold should be very similar to that for the other two species given they all have similar estimated whole genome coverage? I think that something has not worked properly for this part of the analysis with the *F. nigra* assembly and I would urge the authors to try to resolve what has happened here. Also, on lines 183-184 the Authors say, in reference to the HapPy results, “*F. americana* and *F. pennsylvanica* having a secondary peak around twenty, representing diploid sequences” - this is incorrect; the 20x peak represents the haploid peak (where sequences from both haplotypes are already collapsed in the assembly). The Authors should check that they have not generally mixed up the haploid and diploid peaks in their description of these results – the diploid peak (uncollapsed sequences) is the lower coverage peak (on the left in Figure S2a and S2e) and the haploid peak (collapsed sequences) is the higher coverage peak.

Introduction

2. On lines 70-72, the Authors indicate that the putative loss-of-function mutations detected by Kelly et al., (2020) were absent from an “EAB-resistant” *F. pennsylvanica*. However, this was not actually checked for in that study because putative loss-of-function mutations were only examined in the subset of 10 *Fraxinus* taxa that had been included in the initial convergence analyses, which did not include the “EAB-resistant” *F. pennsylvanica* genotype.

3. On lines 145-146, the Authors suggest that “further advancements in sequencing technology” could allow the methods they described to be applied to species with larger genome sizes. I think this is a pretty meaningless statement that doesn’t add anything of value to the reader – if the Authors mean to say that with further improvements the output from a single MinION flow cell might have increased to make that approach applicable to larger genomes as well, then they should say that clearly.

Results and Discussion

4. Lines 154-155: “as calculated using data from Whittemore et al.” - what calculation is being done here, are these not just the genome sizes as reported in Whittemore et al?

5. Lines 234-237: rather than saying that “older events” have been “estimated to be earlier” I think it would be better to say “earlier events” have been “estimated to have younger ages”.

6. Lines 381-384: the Authors suggest that the *F. pennsylvanica* individual they sequenced could potentially be from a population with a history of introgression between *F. pennsylvanica* and *F. americana*. Given that the *F. americana* genome assembly is available and an independent *F. pennsylvanica* assembly is also available, might it not be possible to explicitly test for this, especially

given the fact that *F. pennylvanica* and *F. americana* are apparently not sister species. E.g. see <https://doi.org/10.1093/genetics/iyab173>.

7. Lines 395-396: I don't think it really makes sense to talk about differences in population structure between individuals – suggest rephrasing.

8. Lines 412-414: If it is suggested that there was been widespread hybridisation and ancient introgression within the Oleaceae, wouldn't a network approach be more appropriate for inferring evolutionary relationships? (not that I am suggesting the Authors should necessarily do more analysis here, but perhaps it is worth noting for the future)

9. There are a few spelling errors/typos: line 489 - "contigulous" should be contiguous; line 532 - "Fraxius" should be Fraxinus; line 533 - "and and" needs the second and removing;

Figures

10. Line 685: it doesn't make sense to talk about "paleodemographic analyses for our three long-read Fraxinus species "; suggest rephrasing to something like "paleodemographic analyses for three Fraxinus species based on our long-read assemblies".

Reviewer #2 (Remarks to the Author):

The author has adequately addressed all comments.

We thank the reviewers for their constructive comments. Please refer to our responses to each point below, highlighted in yellow among the referees' comments.

Reviewer #1 (Remarks to the Author):

This is a revised version of a manuscript that I reviewed previously. The authors have made detailed responses to the points raised by all three reviewers and I am satisfied overall with the changes they have made to the text and analyses. There is however one point that I raised previously that I feel still needs further clarification – this is why the coverage for *F. nigra* is some of the analyses is estimated at about half that expected. I also have a few additional minor comments, which are outlined further below.

1. In response to my previous comment on this topic (number 10 in my original review), the authors outline how on the basis of mapping long reads either to the initial assembly for *F. nigra*, or to the haploid assembly, the estimated coverage is c. 21x but that both before and after purging of haplotigs the estimated coverage with HapPy is 11x. However, this does not explain why this is the case. If this species truly had very few haplotigs in the initial assembly and the coverage of the genome is truly at around 21x, as suggested from the read mapping, then we would expect to see a peak at c. 21x from the HapPy coverage analysis, not 11x. Also, it makes no sense that the inferred threshold for coverage between the diploid and haploid parts of the assembly is at 1x for an assembly where it is suggested there is actually 21x coverage – surely this threshold should be very similar to that for the other two species given they all have similar estimated whole genome coverage? I think that something has not worked properly for this part of the analysis with the *F. nigra* assembly and I would urge the authors to try to resolve what has happened here.

We thank the reviewer for bringing up this concern. While the overall depth of coverage of all three genomes is ~22x when taking all contigs into account, *F. nigra* has a few extremely high coverage contigs (contig_184 and contig_7 with 24,085x and 30,199x coverage, respectively). These levels of coverage were an order of magnitude larger than the contigs with the highest depth of coverage in *F. americana* (~4,150x) or *F. pennsylvanica* (~3,330x). When entered in NCBI BLAST, these two *F. nigra* sequences matched most closely with *F. nigra* chloroplast sequences already present in NCBI. When we progressively remove the contigs with the highest coverage from the average depth of coverage calculation ((sum(contig coverage x contig length))/total length of all contigs included), you see that *F. americana* and *F. pennsylvanica* stay around ~22x coverage, but *F. nigra* progressively decreases to 13.8%. Here is a table with average depth of coverage for all three species as progressively remove more contigs with the highest coverage

	Fraxinus americana	Fraxinus nigra	Fraxinus pennsylvanica
average depth of coverage (0-30,199x)	N/A	22.2674	N/A
average depth of coverage (0-10,000x)	24.3127	22.0161	22.7736
average depth of coverage (0-1,000x)	23.6897	20.3995	22.3188
average depth of coverage (0-100x)	22.4415	15.5966	21.7663
average depth of coverage (0-50x)	22.1130	14.3511	21.6378
average depth of coverage (0-25x)	21.5133	13.7727	20.8633

When we revisited the Flye assembly results, Flye outputs some assembly stats and did give an average depth of coverage of 13x for *F. nigra*, 19x for *F. americana*, and 16x for *F. pennsylvanica* due to filtering out reads under their quality threshold. Because of this difference in our *F. nigra* assembly, the plotted coverage peak did not match the overall average coverage of the genome as a whole. Despite our *F. nigra* genome having low coverage for its nuclear sequence, we're highly confident that the genome is reliable due to our syntenic analyses, success with scaffolding (without correction or patching) using a reference, high complete BUSCO scores, and an expected percentage of complete and duplicated BUSCO scores.

Also, on lines 183-184 the Authors say, in reference to the HapPy results, “*F. americana* and *F. pennsylvanica* having a secondary peak around twenty, representing diploid sequences” - this is incorrect; the 20x peak represents the haploid peak (where sequences from both haplotypes are already collapsed in the assembly). The Authors should check that they have not generally mixed up the haploid and diploid peaks in their description of these results – the diploid peak (uncollapsed sequences) is the lower coverage peak (on the left in Figure S2a and S2e) and the haploid peak (collapsed sequences) is the higher coverage peak.

We thank the reviewer for bringing this mistake to our attention. There was no mistake in analysis, just in what we wrote. We should have said that *F. americana* and *F. pennsylvanica* had a haploid peak at ~20x and a diploid peak at ~10x. This has been fixed where it is mentioned throughout the manuscript.

Introduction

2. On lines 70-72, the Authors indicate that the putative loss-of-function mutations detected by Kelly et al., (2020) were absent from an “EAB-resistant” *F. pennsylvanica*. However, this was not actually checked for in that study because putative loss-of-function mutations were only examined in the subset of 10 *Fraxinus* taxa that had been included in the initial convergence analyses, which did not include the “EAB-resistant” *F. pennsylvanica* genotype.

We thank the reviewer for pointing out this mistake. What we wrote was not consistent with what was written in Kelly et al. (2020). The putative EAB-resistant *F. pennsylvanica* was not included in the loss-of-function mutations analysis. We decided to remove the two sentences discussing the susceptible and putatively resistant *F. pennsylvanica* individuals.

3. On lines 145-146, the Authors suggest that “further advancements in sequencing

technology” could allow the methods they described to be applied to species with larger genome sizes. I think this is a pretty meaningless statement that doesn’t add anything of value to the reader – if the Authors mean to say that with further improvements the output from a single MinION flow cell might have increased to make that approach applicable to larger genomes as well, then they should say that clearly.

We agree with this, and it has been made clearer in the main text.

Results and Discussion

4. Lines 154-155: “as calculated using data from Whittemore et al.” - what calculation is being done here, are these not just the genome sizes as reported in Whittemore et al?

Whittemore et al. has an appendix that lists individual 2C genome sizes based on flow cytometry experiments. They don’t directly give mean 2C values for each species. *F. americana* has 105 2C values in picograms (pg), *F. nigra* has 19 2C values in pg, and *F. pennsylvanica* has 111 2C values in pg. I put these values into an excel sheet, took the averages, converted them into megabases (MB) by multiplying the pg value by 0.978, and divided the value in half for the 1C value in MB. I didn’t include the tables because the work was essentially already done by Whittemore et al., but they didn’t publish the exact values.

5. Lines 234-237: rather than saying that “older events” have been “estimated to be earlier” I think it would be better to say “earlier events” have been “estimated to have younger ages”.

This has been updated in the manuscript.

6. Lines 381-384: the Authors suggest that the *F. pennsylvanica* individual they sequenced could potentially be from a population with a history of introgression between *F. pennsylvanica* and *F. americana*. Given that the *F. americana* genome assembly is available and an independent *F. pennsylvanica* assembly is also available, might it not be possible to explicitly test for this, especially given the fact that *F. pennsylvanica* and *F. americana* are apparently not sister species. E.g. see <https://doi.org/10.1093/genetics/iyab173>.

We thank the reviewer for this suggestion. We agree that this can be an interesting analysis to perform, but we believe it’s beyond the scope and purpose of this study.

7. Lines 395-396: I don’t think it really makes sense to talk about differences in population structure between individuals – suggest rephrasing.

This has been updated in the manuscript.

8. Lines 412-414: If it is suggested that there was been widespread hybridisation and ancient introgression within the Oleaceae, wouldn’t a network approach be more appropriate for inferring evolutionary relationships? (not that I am suggesting the

Authors should necessarily do more analysis here, but perhaps it is worth noting for the future)

While we agree that a network would more accurately describe the relationships between the species in our tree, a network approach was recently done by Dong et al. (2022). For our purposes, the bifurcating tree generated by Orthofinder was sufficient. Even though our tree swapped *Jasminum* and *Forsythia* compared with Dong et al. (2022), for example, this wasn't an issue for this analysis since *Jasminum* and *Forsythia* have the same ploidy level. This bifurcating OrthoFinder tree was sufficient to provide additional evidence for the history of ploidy events within the Oleaceae. A sentence was added about a network approach being more accurate than a bifurcating tree.

9. There are a few spelling errors/typos: line 489 - "contigulous" should be contiguous; line 532 - "Fraxius" should be Fraxinus; line 533 - "and and" needs the second and removing;

This has been corrected in the manuscript.

Figures

10. Line 685: it doesn't make sense to talk about "paleodemographic analyses for our three long-read Fraxinus species "; suggest rephrasing to something like "paleodemographic analyses for three Fraxinus species based on our long-read assemblies".

This has been updated in the manuscript.

Reviewer #2 (Remarks to the Author):

The author has adequately addressed all comments.

REVIEWERS' COMMENTS:

Reviewer #1 (Remarks to the Author):

This is a revised version of a manuscript that I have reviewed twice previously. The authors have responded to the points I raised and I am satisfied overall with the changes they have made and have no further comments.